# Plain Transformers are Surprisingly Powerful Link Predictors

Quang Truong [1]   Yu Song [1]   Donald Loveland [2]   Mingxuan Ju [2]   Tong Zhao [2]   Neil Shah [2]   Jiliang Tang [1]

## Abstract

Link prediction is a core challenge in graph machine learning, demanding models that capture rich and complex topological dependencies. While Graph Neural Networks (GNNs) are the standard solution, state-of-the-art pipelines often rely on explicit structural heuristics or memory-intensive node embeddings—approaches that struggle to generalize or scale to massive graphs. Emerging Graph Transformers (GTs) offer a potential alternative but often incur significant overhead due to complex structural encodings, hindering their applications to large-scale link prediction. We challenge these sophisticated paradigms with PENCIL, an encoder-only plain Transformer that replaces hand-crafted priors with attention over sampled local subgraphs, retaining the scalability and hardware efficiency of standard Transformers. Through experimental and theoretical analysis, we show that PENCIL extracts richer structural signals than GNNs, implicitly generalizing a broad class of heuristics and subgraph-based expressivity. Empirically, PENCIL outperforms heuristic-informed GNNs and is far more parameter-efficient than ID-embedding–based alternatives, while remaining competitive across diverse benchmarks—even without node features. Our results challenge the prevailing reliance on complex engineering techniques, demonstrating that simple design choices are potentially sufficient to achieve the same capabilities. Our code is publicly available at https://github.com/quang-truong/pencil.

[1]Department of Computer Science and Engineering, Michigan State University, East Lansing, MI, USA. [2]Snap Inc., Bellevue, WA, USA. Correspondence to: Quang Truong <truongc4@msu.edu>.

*Proceedings of the 43rd International Conference on Machine Learning*, Seoul, South Korea. PMLR 306, 2026. Copyright 2026 by the author(s).

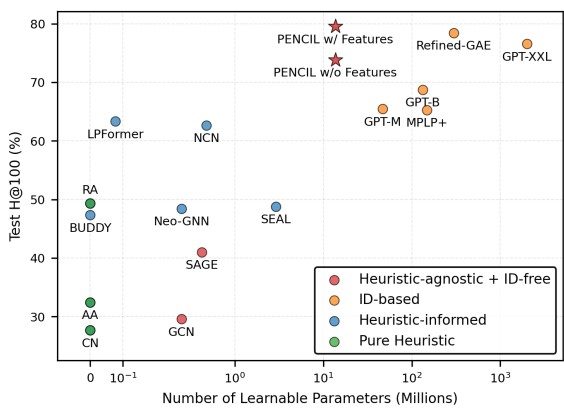

*Figure 1.* Parameter efficiency vs. performance on `ogbl-ppa`. PENCIL (★) achieves state-of-the-art performance without node IDs or handcrafted heuristics, using orders of magnitude fewer parameters than leading ID-based methods.

## 1. Introduction

Link prediction, the task of inferring missing edges between node pairs in graph-structured data, is a fundamental primitive with applications spanning recommendation systems (Huang et al., 2005) and drug discovery (Abbas et al., 2021). Given its centrality in graph learning, Graph Neural Networks (GNNs) have become the de facto standard for this challenge. However, standard Message-Passing Neural Networks (MPNNs), the primary engine for most GNN-based encoders, are inherently limited by their node-centric aggregation, which often renders structurally distinct node pairs indistinguishable if the individual nodes possess symmetric local neighborhoods (Zhang et al., 2021). Therefore, state-of-the-art architectures typically integrate diverse structural signals, ranging from local heuristics (Wang et al., 2024; Yun et al., 2021; Chamberlain et al., 2023) and pairwise encodings (Li et al., 2020; Zhang & Chen, 2018) to global graph heuristics (Shomer et al., 2024) and learnable per-node embeddings (Ma et al., 2025b; Dong et al., 2024), through either implicit or explicit architectural modifications.

While these augmented architectures address expressiveness, their design choices often compromise scalability and complicate deployment. First, reliance on learned per-node embeddings (ID-based) or precomputed global features of-

ten imposes a static dependency on the initial graph state, necessitating expensive retraining or recomputation to incorporate new nodes (Fey et al., 2024; Longa et al., 2023). Second, at web scale, methods requiring whole-graph passes or the frequent refresh of globally materialized caches create prohibitive memory and update overhead; this necessitates predictors capable of operating on fixed-budget, mini-batch sampled neighborhoods (Zhao et al., 2025b; Fey et al., 2024). Third, as noted by (Li et al., 2023), reliance on simple structural cues like common neighbors often results in inflated performance metrics, allowing models to succeed through exploitation rather than by learning generalizable patterns. These limitations collectively motivate a shift toward link predictors designed for realistic deployment scenarios: ID-free, mini-batch efficiency, and structural robustness with minimal reliance on global preprocessing.

To address the limitations of GNNs, recent research has pivoted toward Graph Transformers (GTs). By replacing restricted neighborhood aggregation with all-to-all attention, GTs can be theoretically a universal function approximator under appropriate structural conditioning (Müller et al., 2023). Yet, there is *no consensus on how to bake graph structure into Transformers*. Specifically, researchers typically integrate positional (PE) and structural encodings (SE) derived from spectral signals (Rampášek et al., 2022; Kreuzer et al., 2021; Dwivedi & Bresson, 2021), random walks (Chen et al., 2025; Kim et al., 2025), graph heuristics (Ying et al., 2021), or GNNs (Jain et al., 2021; Dwivedi et al., 2025). However, while these encodings demonstrate proven success on graph-level tasks, these components are often infeasible for link prediction because they are processed offline or require entire graph structure, thus violating the constraints set above. The field also faces significant architectural heterogeneity, as there is no canonical way to adapt the *plain Transformer* (Vaswani et al., 2017)—originally designed for sequential data—to irregular graph structures (Ma et al., 2025a). Current GT designs often necessitate specialized graph-specific attention kernels (Ma et al., 2023; Shomer et al., 2024), or sophisticated input processing pipelines that introduce substantial computational overhead (Chen et al., 2022; Zhao et al., 2025a). Such fundamental departures from standard architectures prevent these models from fully leveraging highly optimized hardware accelerator toolchains, such as efficient attention kernel (Dao et al., 2022). Consequently, there is a clear vacancy for a Transformer-based link predictor that is expressive while adhering to the above deployment constraints.

In this work, we propose **Plain ENCoder for Inferring Links** (PENCIL), a Transformer-based link predictor that utilizes a standard BERT-style encoder (Devlin et al., 2019) for full compatibility with modern hardware. In particular, PENCIL evaluates each candidate link from a fixed-budget, sampled link-centric local neighborhood—without hand-crafted heuristic features—enabling straightforward mini-batching while avoiding reliance on globally materialized caches. Notably, PENCIL **does not need costly offline computation of PEs/SEs**, which differentiates it from existing GTs. To our knowledge, this work provides the first demonstration that plain Transformers serve as highly effective link predictors under strict deployment constraints. Furthermore, we provide a formal theoretical analysis that characterizes the mechanisms underlying this performance and situates the model's expressive power relative to established link prediction paradigms. Main contributions are:

**Transformer-based Link Predictor.** We introduce a Transformer architecture that learns expressive representations from *only* sampled local subgraphs. As highlighted in Figure 1, it exceeds the performance of others while using $22\times$ to $146\times$ fewer learnable parameters than the next best competitors. Notably, PENCIL achieves a substantial advancement in training efficiency on large-scale datasets, requiring between $6.7\times$ and $40\times$ fewer epochs to converge than pure GNN architectures. Our results indicate that local, sampled contexts provide ample signal for high performance, highlighting a potential over-reliance in existing models on auxiliary components over inherent structural information.

**Theoretical Unification.** We provide a theoretical analysis connecting PENCIL to established link prediction paradigms. We show that PENCIL inherently formulates many traditional structural heuristics by design, achieving the expressivity of subgraph-based predictors like SEAL (Zhang & Chen, 2018) without explicit hard-coding distance-based labels.

**Data Insights.** We find that, on several standard benchmarks, PENCIL can achieve strong performance using structural information alone. This suggests that node features can be weakly informative relative to local structure and may provide limited marginal gains on many datasets.

## 2. Preliminaries

In this section, we will introduce notations and concepts that will be used throughout the paper.

**Notations.** Let $G = (\mathbf{A}, \mathbf{X})$ denote an (undirected) graph with adjacency matrix $\mathbf{A} \in \{0, 1\}^{M \times M}$ and node-feature matrix $\mathbf{X} \in \mathbb{R}^{M \times f}$, where $M$ is the number of nodes and $f$ is the feature dimension. Let degree matrix be $\mathbf{D}$. The node set is $\mathcal{V} = \{1, \ldots, M\}$ and the edge set is $\mathcal{E} = \{(i, j) \in \mathcal{V} \times \mathcal{V} : \mathbf{A}_{ij} = 1\}$. For $v \in \mathcal{V}$, we define the neighborhood as $\mathcal{N}(v) = \{u \in \mathcal{V} : \mathbf{A}_{uv} = 1\}$. For any matrix square $\mathbf{M}$, its $i$-th row is denoted by $\mathbf{m}_i = \mathbf{M}_i$. Let $\mathbf{P}_\pi$ denote permutation matrix of node relabeling $\pi$.

**Link Prediction.** Given an observed graph $G = (\mathbf{A}, \mathbf{X})$ with edge set $\mathcal{E}$, the link prediction task assigns a score

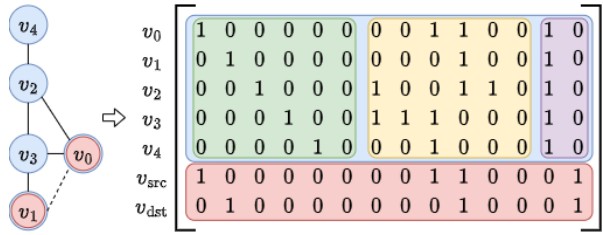

*Figure 2.* Visualization of the input encoding scheme for a sampled subgraph revolving around a query link (dashed), with the number of nodes $N = 5$ and the sampling budget $N_{\max} = 6$. Blue block corresponds to the sampled nodes (the context set). Green block contains one-hot identifiers of nodes, yellow block contains nodes' adjacency row, purple block contains role flags, and red block contains task tokens. Context-node ordering is permutable; only $v_{\text{src}}$ and $v_{\text{dst}}$ are fixed to $v_0$ and $v_1$.

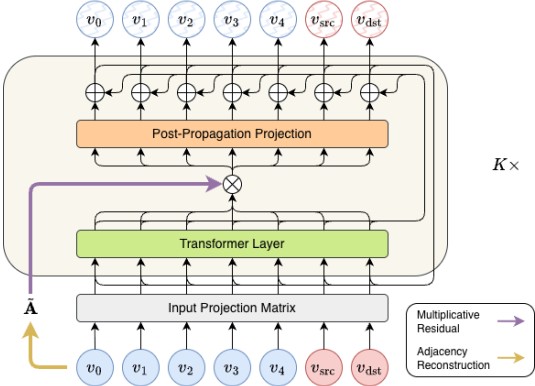

*Figure 3.* PENCIL architecture. Node features can be optionally used as discussed in Section 3.2, but we omit them from the figure for clarity.

to a candidate node pair $(u, v) \in \mathcal{V} \times \mathcal{V}$ indicating the likelihood that an edge exists between $u$ and $v$. Under a temporal split, $\mathcal{E}$ contains edges observed up to a cutoff time, while evaluation targets future edges; thus, the edges to be predicted are not necessarily a subset of $\mathcal{E}$.

**Pairwise Heuristics.** A pairwise heuristic is a scoring function $h$ that, given the observed graph $G$, maps a candidate node pair $(u, v) \in \mathcal{V} \times \mathcal{V}$ to a real-valued score. Candidate pairs are then ranked by this score to estimate the likelihood that an edge exists between $u$ and $v$. Common pairwise heuristics are summarized in Appendix C.1.

## 3. Plain Transformers As Link Predictors

In this section, we describe graph encoding scheme, the architecture of PENCIL, and its use for estimating pairwise heuristics relative to classic GNNs.

### 3.1. Graph Encoding Scheme

We encode each sampled subgraph around a candidate pair $(v_{\text{src}}, v_{\text{dst}})$ using an encoding scheme similar to node-adjacency tokenization (Yehudai et al., 2026). As illustrated in Figure 2, we assign indices to the $N$ nodes in the extracted subgraph (with $N \le N_{\max}$ where $N_{\max}$ is the sampling budget) by fixing $v_{\text{src}}$ to index 0 and $v_{\text{dst}}$ to index 1, while assigning the remaining nodes uniformly at random to indices $2, \dots, N - 1$. For the first $N$ *context* nodes (blue block), we concatenate three parts: (i) a one-hot identifier of the node in the chosen subgraph order, padded to length $N_{\max}$ (green block); (ii) the node's adjacency indicator row within the extracted subgraph, also padded to length $N_{\max}$ (yellow block); and (iii) a two-bit role flag indicating whether the node is a context node or a task node (purple block), set to $[1, 0]$ for context nodes. We then append two *task* nodes corresponding to the endpoints $v_{\text{src}}$ and $v_{\text{dst}}$ (bottom two red rows): each task node copies the same one-hot and adjacency parts from its corresponding context node representation, but flips the role flag to $[0, 1]$. Since 'node' and 'token' are referring to the same entity, we use them interchangeably from now on.

### 3.2. Architecture Design

We begin by outlining the subgraph batching, which will clarify the tensor dimensions used later. Each mini-batch contains $B$ sampled subgraphs; letting $N_b$ be the number of context nodes in subgraph $b$, we set $N_B = \max_{b \in [B]} N_b$ and pad all samples to this length. The model processes a length-$(N_B + 2)$ token sequence consisting of $N_B$ context tokens (sampled subgraph nodes, with padding as needed) and two task tokens (for $v_{\text{src}}$ and $v_{\text{dst}}$). By construction, $N_B \le N_{\max}$, where $N_{\max}$ is the global sampling budget. It is worth noting that there are two levels of padding: padding to $N_{\max}$ is feature-level padding, used for the one-hot identifier and adjacency fields, whereas padding to $N_B + 2$ is batch-level sequence padding, accounting for the two appended task tokens. While one can pad every sequence to $N_{\max} + 2$, it is more efficient to pad to $N_B + 2$, avoiding unnecessary computation when sampled subgraphs are smaller than $N_{\max}$.

Figure 3 illustrates PENCIL, a model composed of $K$ stacked blocks of identical form that couple bidirectional self-attention with one-hop propagation. We omit the default sequential PE, so the Transformer encoder is permutation equivariant with respect to reordering the input tokens.

**Input Projection.** The tokenized input $\tilde{\mathbf{X}} \in \mathbb{R}^{B \times (N_B + 2) \times (2N_{\max} + 2)}$ (see Figure 2) and optional node features $\mathbf{X} \in \mathbb{R}^{B \times (N_B + 2) \times f}$ are projected into a shared hidden dimension $d$ to form $\mathbf{H}^{(0)}$:

$$\mathbf{H}^{(0)} = \tilde{\mathbf{X}} \mathbf{W}_0 + \mathbf{G}(\mathbf{X}) \in \mathbb{R}^{B \times (N_B + 2) \times d}, \quad (1)$$

where $\mathbf{W}_0 \in \mathbb{R}^{(2N_{\max}+2) \times d}$ is an orthogonally initialized projection matrix and $\mathbf{G}(\cdot)$ is a learnable feature encoder. Although random initialization often yields approximately orthogonal vectors in high dimensions, it does not guarantee an orthonormal or norm-preserving projection. We therefore use orthogonal initialization (Saxe et al., 2014) with gain 1 to obtain a better-conditioned initial projection.

**Adjacency Reconstruction.** A key feature of PENCIL is that the subgraph adjacency matrix $\tilde{\mathbf{A}}$ is not provided as a separate input but is recovered directly from the token encoding $\tilde{\mathbf{X}}$. Since each token includes its adjacency-indicator row (see Figure 2), $\tilde{\mathbf{A}}$ is constructed by slicing the corresponding columns from $\tilde{\mathbf{X}}$. While normalized variants such as $\mathbf{D}^{-1}\tilde{\mathbf{A}}$ or $\mathbf{D}^{-1/2}\tilde{\mathbf{A}}\mathbf{D}^{-1/2}$ are applicable, we retain the raw adjacency $\tilde{\mathbf{A}}$ for notational simplicity. Details regarding the adjacency reconstruction are discussed in Appendix A.

**Multiplicative Residual.** For each layer $k$, PENCIL applies a bidirectional Transformer block followed by a residual term:

$$\mathbf{Z}^{(k)} = \mathbf{T}_k\Big(\mathbf{H}^{(k-1)}\Big) \in \mathbb{R}^{(N_B+2) \times d}, \qquad (2)$$

$$\mathbf{H}^{(k)} = \mathbf{Z}^{(k)} + \mathbf{P}_k\Big(\tilde{\mathbf{A}}\mathbf{Z}^{(k)}\Big) \in \mathbb{R}^{(N_B+2) \times d}, \qquad (3)$$

where $\mathbf{T}_k$ is a Transformer layer and $\mathbf{P}_k$ is a learnable row-wise post-propagation projection applied to $\tilde{\mathbf{A}}\mathbf{Z}^{(k)}$. We refer to the second term as a **multiplicative residual** because it adds an explicit matrix-multiplication branch $\mathbf{P}_k(\tilde{\mathbf{A}}\mathbf{Z}^{(k)})$ on top of the attention output, where $\tilde{\mathbf{A}}$ is reconstructed from the input encoding $\tilde{\mathbf{X}}$. Therefore, if self-attention is replaced with the identity map $\mathbf{Z}^{(k)} = \mathbf{H}^{(k-1)}$, the update reduces to a residual message-passing layer over $\tilde{\mathbf{A}}$, with node features induced by the graph encoding scheme in Section 3.1. As the residual is computed outside the self-attention module, resulting in full compatibility with efficient attention techniques. Finally, self-attention is computed only among non-padded tokens from the same sampled subgraph.

The link logit is obtained by concatenating the final representations of $v_{\text{src}}$ and $v_{\text{dst}}$, followed by a linear projection to a scalar. The model is trained with the binary cross-entropy (BCE) loss.

### 3.3. Estimating Pairwise Heuristics with PENCIL

Unlike link prediction, which depends on many factors beyond topology, pairwise heuristics are determined by graph structure alone. Furthermore, prior work has established theoretical links between these heuristics and link prediction performance, explaining why they can already serve as strong baselines (Sarkar et al., 2010). Pairwise heuristics therefore provide a clean benchmark for assessing how well Transformers can recover fundamental structural sig-

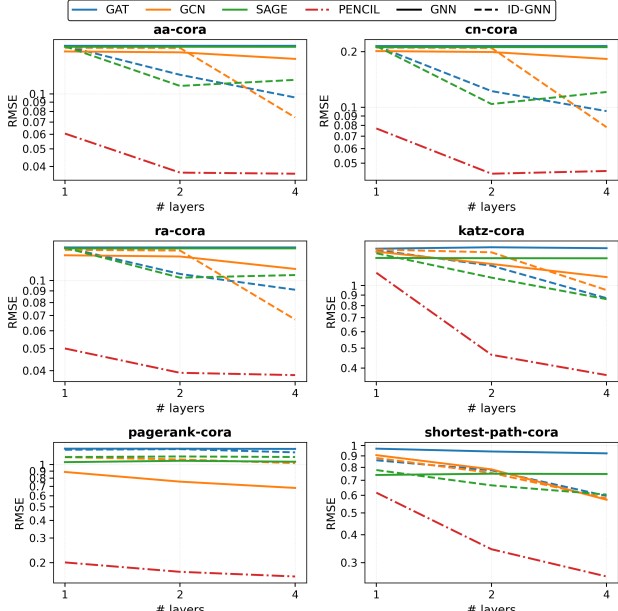

*Figure 4.* RMSE of PENCIL and other GNNs for estimating pairwise heuristics on the `cora` dataset.

nals and where their limitations may lie. In this section, we evaluate PENCIL on a *pairwise heuristic estimation* task and compare against standard GNN baselines.

For each candidate pair $(u, v)$, we compute a heuristic score on the *full* graph and use it as a scalar regression target. We include both *local* heuristics—Common Neighbors (CN), Adamic–Adar (AA), and Resource Allocation (RA)—and *global* heuristics that aggregate multi-hop or graph-wide structure, namely Katz index (Katz, 1953), shortest-path distance (SPD), and a PageRank-based pair score $\mathrm{PR}(u, v) = p_u p_v$ (Brin & Page, 1998). In contrast, during training and evaluation all models are restricted to an induced subgraph *centered around* $(u, v)$ produced by the sampling procedure, and are not given access to the full graph. The model must therefore approximate full-graph statistics using only restricted local subgraphs. Our aim is to quantify, under the sampling constraints, how accurately different architectures can recover these heuristic signals. Figure 4 reports RMSE on `cora`; additional setup details and `citeseer` results are provided in Appendix C.1.

Based on Figure 4, we observe three distinct trends. First, for local heuristics (AA, CN, RA), standard GNNs show negligible improvement with depth. In contrast, ID-GNNs significantly reduce error within the first two layers before performance saturates. Second, for long-range heuristics (Katz index, SPD), increased depth yields clearer benefits, most notably for PENCIL, where RMSE steadily decreases as the model deepens. Third, PageRank proves to be the most challenging target for MPNN baselines; neither ID

augmentation nor added depth bridges the performance gap, whereas PENCIL achieves substantially lower RMSE across all depths. Notably, the same observation regarding GNNs' subpar performance is reported in (Liang et al., 2024), where they suggest that GNNs need learnable node embeddings to learn pairwise heuristics.

Overall, PENCIL surpasses all GNN and ID-GNN models even with a single layer, achieves low error on local metrics with just two layers, and effectively leverages depth to capture complex properties like Katz index, SPD, and PageRank. Similar trends are observed on `citeseer` (see Appendix C.1). Collectively, these results indicate that under identical sampling constraints, PENCIL extracts structural information more effectively than MPNNs, validating its potential as a robust link predictor. Furthermore, many state-of-the-art methods incorporate these heuristics as explicit input signals (Wang et al., 2024; Yun et al., 2021; Dong et al., 2024); in contrast, our results show that a Transformer can estimate them directly from the input graph, without hard-coding them as features.

## 4. Theoretical Analysis

In this section, we would like to provide theoretical analysis on PENCIL, particularly how it can work on link prediction tasks, its surprising capability on pairwise heuristic estimation, and its expressive power with respect to other link prediction models.

### 4.1. Distributional Permutation Invariance

In graph learning, it is standard to require symmetry with respect to node relabelings. Concretely, intermediate representations should be *permutation equivariant*, whereas task-level predictions should be *permutation invariant*—meaning the output remains unchanged under arbitrary node permutations (Zaheer et al., 2017; Maron et al., 2019; Bodnar et al., 2021; Truong & Chin, 2024).

PENCIL is not *deterministically* invariant to node relabeling. This is because adjacency-row tokenization relies on a randomized index assignment: it fixes the queried endpoints to canonical positions and assigns random indices to the remaining nodes. While this breaks deterministic invariance for a fixed sample, it does not violate the underlying symmetry required for graph tasks. We show that the induced predictor is *permutation invariant in distribution* over the randomness of this assignment.

**Theorem 4.1.** *Let $f$ be any deterministic measurable function. Define a randomized predictor $S(\mathbf{A}; u, v) := f(\mathbf{P}_\rho \mathbf{A} \mathbf{P}_\rho^\top)$, where $\rho$ is drawn uniformly from the set of permutations satisfying $\rho(u) = 0$ and $\rho(v) = 1$. Then for*

*any node relabeling $\pi$, the output distribution is unchanged:*

$$S(\mathbf{A}; u, v) \overset{d}{=} S(\mathbf{A}'; u', v'),$$

*where $\mathbf{A}' = \mathbf{P}_\pi \mathbf{A} \mathbf{P}_\pi^\top$, $u' = \pi(u)$, and $v' = \pi(v)$.*

Detailed proof is included in Appendix B.1. Theorem 4.1 formalizes that, although our encoding is not deterministically invariant for a fixed canonicalization, the resulting randomized predictor remains a valid graph function: relabeling the graph and the query pair leaves the predictor unchanged *in distribution*.

Averaging over multiple independent canonicalizations can reduce the variance induced by random index assignments and better approximate the invariant predictor, but it requires additional forward passes at inference time. In practice, a single labeled subgraph per forward pass already achieves strong performance, as shown in Section 5. Appendix C.5 further shows that using multiple labeled subgraphs with DeepSets-style pooling (Zaheer et al., 2017) does not yield clear gains, supporting the efficiency of our design.

### 4.2. What Makes PENCIL a Strong Link Predictor?

In this section, we draw connections between PENCIL and prior works on link prediction, namely NBFNet (Zhu et al., 2021) and MPLP (Dong et al., 2024), to provide theoretical evidence for its surprising capability on pairwise heuristic estimation demonstrated in Section 3.3.

NBFNet (Zhu et al., 2021) is a source-conditioned message-passing framework (You et al., 2021) for link prediction that propagates information from a query source node and scores links by reading out the representations at candidate destination nodes. It can be viewed as a learnable relaxation of the generalized Bellman–Ford algorithm that recovers many path-based heuristics; details are provided in Appendix B.2. Next, we show that PENCIL can be degenerated to an NBFNet-style propagation model, subsequently leading to the following corollary.

**Proposition 4.2.** *There exists a parameter setting of PENCIL under which its layerwise update reduces to a source-conditioned MPNN, with readout at the canonical destination token $v_1$.*

**Corollary 4.3.** *Under suitable parameter settings and operator choices, PENCIL can realize a broad class of classical path-based link prediction scores and graph algorithms, including Katz index, Personalized PageRank, SPD, widest path, and most reliable path.*

Proofs are provided in Appendix B.3 and Appendix B.4. Next, we show that PENCIL can also estimate local pairwise heuristics. First, we have the following remark, which is an informal statement of Theorem B.3 and Theorem B.4 in Appendix B.5, proven in (Dong et al., 2024).

*Remark* 4.4. If the initial node features are random, zero-mean, and (in expectation) unit-norm, then sum-aggregation message passing turns dot products into counts of shared connectivity patterns.

The above remark suggests that sum-aggregation MPNNs can estimate local heuristics, depending on the initial node representations—namely, zero mean and unit norm in expectation. Importantly, these conditions are sufficient but not restrictive in practice: they are already satisfied (or closely approximated) by common ways of initializing learnable node embeddings, such as zero-mean random initialization with appropriate scaling, and in particular orthogonal initialization. Motivated by this perspective, (Dong et al., 2024) studies quasi-orthogonal node signatures, while (Ma et al., 2025b) initializes learnable node embeddings with orthogonal vectors. We now show that PENCIL can be configured to satisfy the same assumptions, and therefore inherits the same unbiased estimation property.

**Proposition 4.5.** *There exists a parameter setting of PENCIL under which it can estimate heuristics stated in Theorem B.3 and Theorem B.4.*

Proof is provided in Appendix B.6. A key distinction from MPLP (Dong et al., 2024) and Refined-GAE (Ma et al., 2025b) is where these vectors live. Those methods conceptually maintain a (quasi-)orthogonal vector per node, whereas in PENCIL the vectors are induced by the input projection over the *sampled* token set. Consequently, the number of vectors that must be simultaneously well-separated is bounded by the sampling budget (i.e., $N_{\max} \ll |\mathcal{V}|$). Since the accuracy of the above unbiased estimators depends on orthogonality of vectors, we quantify this property via mutual coherence (Donoho & Huo, 2001) and the Welch bound (Welch, 1974).

**Definition 4.6** (Mutual coherence). Let $\mathcal{F} = \{\mathbf{r}_1, \ldots, \mathbf{r}_N\}$ be a set of $N$ unit-norm vectors in $\mathbb{R}^d$. The *mutual coherence* of $\mathcal{F}$, denoted $\mu(\mathcal{F})$, is defined as the maximum absolute inner product between any distinct pair of vectors in the set:

$$\mu(\mathcal{F}) \triangleq \max_{1 \leq i \neq j \leq N} |\langle \mathbf{r}_i, \mathbf{r}_j \rangle|. \quad (4)$$

**Definition 4.7** (The Welch bound). For any set size $N$ and dimension $d$ (where $N > d$), the *Welch bound* $W(N, d)$ defines the theoretical lower bound on the mutual coherence achievable by any set of vectors:

$$\mu(\mathcal{F}) \geq W(N, d) \triangleq \sqrt{\frac{N - d}{d(N - 1)}}. \quad (5)$$

Mutual coherence captures the maximum correlation within a set of unit vectors, quantifying the set's deviation from orthogonality. The Welch bound lower-bounds this quantity as a function of the set size $N$ and dimension $d$, yielding a

fundamental limit on how well-separated $N$ vectors in $\mathbb{R}^d$ can be. We have the following proposition.

**Proposition 4.8.** *For a fixed dimension $d \geq 2$, the Welch bound $W(N, d)$ is strictly increasing as a function of the set size $N$ for all $N > d$.*

Proof is provided in Appendix B.7. This monotonicity reveals an inherent benefit of operating with fewer vectors: for fixed $d$, the smallest achievable mutual coherence increases with $N$, meaning that it becomes increasingly difficult to keep all pairwise correlations small as the set grows. Since PENCIL only needs to keep $N_{\max}$ vectors—rather than maintaining $|\mathcal{V}|$ well-separated node vectors globally—it operates under a strictly weaker coherence constraint. In particular, when $N_{\max} \leq d$, perfectly orthogonal vectors are feasible, which can arise in very large but sparse graphs. In contrast, maintaining strictly orthogonal vectors at the scale of $|\mathcal{V}|$ is infeasible because $|\mathcal{V}| \gg d$.

### 4.3. Expressive Power of Plain Transformers on Link Prediction

Beyond heuristic estimation, PENCIL's expressivity can be probed on highly symmetric graphs by asking whether it distinguishes candidate links whose endpoints are automorphic (structurally interchangeable). Standard node-embedding pipelines often collapse automorphic nodes—and thus such links—to identical representations (Chamberlain et al., 2023), while prior work breaks this symmetry via node labeling (Zhang & Chen, 2018; Zhang et al., 2021).

PENCIL is imbued with a native symmetry-breaking mechanism: the input projection assigns random vectors to tokens, inducing a non-deterministic node labeling. Since each prediction is computed on a sampled subgraph of size at most $N_{\max}$, only finitely many labeled versions of that subgraph arise. This places PENCIL naturally in the family of subgraph-based GNNs (Murphy et al., 2019; Chen et al., 2020; Zhou et al., 2023; Zhao et al., 2022b;a). Moreover, equipping PENCIL with local relational pooling (LRP) (Chen et al., 2020)—i.e., pooling representations across these labeled subgraphs—allows us to evaluate its expressivity within the $k_\phi$-$k_\rho$-$m$ framework (Lachi et al., 2026) and to compare it formally against other paradigms. We leave details about the framework, proof of the following theorem, and further discussion in Appendix B.8.

**Theorem 4.9.** *Under the $k_\phi$-$k_\rho$-$m$ framework, for suitable parameter settings, PENCIL with LRP is not less expressive than SEAL under the same sampling constraint.*

Importantly, (Zhou et al., 2023) prove that labeling more nodes in the subgraph yields strictly stronger expressiveness; thus, increasing the sampling budget $N_{\max}$ directly strengthens PENCIL's expressive power.

# 5. Experimental Results

This section highlights findings: (1) a plain Transformer's surprisingly strong performance on real-world datasets and (2) the impact of model depth under fixed sampling constraints. Further investigations on multiplicative residual, input projection matrix initialization, and computational complexity are included in Appendix C.3, Appendix C.4, and Appendix C.7, respectively.

Regarding experiment settings, we repeat each experiment with five random seeds on the Planetoid benchmarks (`cora`, `citeseer`, and `pubmed`) (Yang et al., 2016) and three random seeds on the OGB link prediction datasets (`ogbl-citation2`, `ogbl-ppa`, `ogbl-ddi`, and `ogbl-collab`) (Hu et al., 2020). Except to the main results, all experiments are conducted without node features. Additionally, `ogbl-ddi` does not have node features. Implementation information is detailed in Appendix C.

## 5.1. Main Results

Table 1 and Table 2 compare PENCIL with a broad set of baselines, including GCN (Kipf & Welling, 2017), SAGE (Hamilton et al., 2017), NBFNet (Zhu et al., 2021), SEAL (Zhang & Chen, 2018), Neo-GNN (Yun et al., 2021), BUDDY (Chamberlain et al., 2023), NCN/NCNC (Wang et al., 2024), LPFormer (Shomer et al., 2024), MPLP+ (Dong et al., 2024), and Refined-GAE (Ma et al., 2025b). In particular, Table 1 reports results under the standard single-split protocol: for the Planetoid datasets we use the fixed split from (Li et al., 2023), and for the OGB datasets we use the official splits from (Hu et al., 2020). We omit `ogbl-ddi` under the original setting due to the low validation–test correlation noted in (Li et al., 2023; Shomer et al., 2024). Table 2 reports results under the HeaRT evaluation protocol (Li et al., 2023), in which negative links are curated per positive test example. The two settings share the same training set and differ only in how negatives are constructed for evaluation. We report the dataset-appropriate metrics recommended by (Li et al., 2023) and (Hu et al., 2020). Baseline results are taken from (Li et al., 2023), while results for LPFormer, MPLP+, and Refined-GAE are quoted as reported in their respective papers. All remaining experimental details are provided in Appendix C.2.

**Performance.** While heuristic-informed and ID-based baselines typically dominate, PENCIL challenges this paradigm by delivering robust performance without such priors. In the original setting, PENCIL achieves state-of-the-art results on `cora` and `ogbl-ppa`; under HeaRT, it secures top scores on `ogbl-ppa` and `ogbl-ddi`. Furthermore, PENCIL demonstrates exceptional stability, consistently exhibiting lower standard deviations than baselines, most notably on `ogbl-ppa`, where its variance is orders of magnitude smaller ($\pm 0.07$) than that of competing methods. Beyond accuracy and stability, PENCIL is highly training-efficient on large-scale benchmarks: it converges on `ogbl-citation2`, `ogbl-ddi`, and `ogbl-ppa` in just 0.5, 8, and 15 epochs, respectively. This is in sharp contrast to pure GNNs, which often require orders of magnitude more training (20–100+ epochs) for the same datasets as reported by (Li et al., 2023). This rapid convergence, however, is most pronounced in the data-rich regime. While PENCIL learns quickly from large datasets, it exhibits lower statistical efficiency on small-scale tasks, where it may underperform leading methods. Consequently, the model fails to achieve the same statistical efficiency as seen with larger datasets. This behavior is consistent with the data-intensive requirements of Transformers when recovering graph algorithms (Sanford et al., 2024) and the "data hunger" of Vision Transformers (ViT) in Computer Vision, which stems from a lack of translation equivariance (Dosovitskiy et al., 2021). In both cases, the absence of hard-coded inductive biases demands a larger volume of data for generalization. See Appendix C.2 for additional training details.

**Structural Sufficiency.** Our results indicate that node features are not universally beneficial and can introduce volatility. On Planetoid, specifically `cora`, adding features degrades performance and significantly inflates variance. In contrast, features provide necessary signal on OGB datasets, yielding major gains on `ogbl-ppa`. This distinct behavior underscores the complex interplay between feature and structure (Coupette et al., 2025; Castellana & Errica, 2023). Given its exceptional performance using structure alone as reported in Table 1 and Table 2, PENCIL potentially serves as a critical baseline for this research direction, capable of effectively isolating structural contributions where other models cannot. For further experiments on synthetic graphs with controlled structural and node-feature signals, see Appendix C.6.

## 5.2. Effects of Model Depth

Standard MPNNs struggle to exploit depth since their receptive field is inherently coupled to graph connectivity, which induces oversmoothing and oversquashing as aggregation expands (Li et al., 2018; Oono & Suzuki, 2020; Alon & Yahav, 2021). Conversely, PENCIL effectively harnesses deeper architectures; its utilization of token-level self-attention facilitates dense communication within subgraph instances, mitigating the topological bottlenecks associated with oversquashing (Topping et al., 2022). As illustrated in Figure 5, increasing model depth yields consistent improvements in Hits@50 and Hits@100 across the `cora`, `pubmed`, and `ogbl-collab` datasets, exhibiting only mild saturation at greater depths. This behavior is markedly different from the MPNN trends documented in Figure 3 of (Ma et al., 2025b), where performance typically peaks at shallow depths before dropping sharply as

*Table 1.* Results on the original benchmark datasets. N/A means results are not available. Colored results indicate the 1st, 2nd, and 3rd-best performance in the corresponding metric. Category markers: ● Heuristic-informed, ● ID-based, and ● Heuristic-agnostic + ID-free.

| Type | Model | cora | citeseer | pubmed | ogbl-collab | ogbl-ppa | ogbl-citation2 |
|---|---|---|---|---|---|---|---|
| | | MRR | MRR | MRR | H@50 | H@100 | MRR |
| ● | SEAL | 26.69 ± 5.89 | 39.36 ± 4.99 | 38.06 ± 5.18 | 63.37 ± 0.69 | 48.80 ± 5.61 | 86.93 ± 0.43 |
| | Neo-GNN | 22.65 ± 2.60 | 53.97 ± 5.88 | 31.45 ± 3.17 | 66.13 ± 0.61 | 48.45 ± 1.01 | 83.54 ± 0.32 |
| | BUDDY | 26.40 ± 4.40 | 59.48 ± 8.96 | 23.98 ± 5.11 | 64.59 ± 0.46 | 47.33 ± 1.96 | 87.86 ± 0.18 |
| | NCN | 32.93 ± 3.80 | 54.97 ± 6.03 | 35.65 ± 4.60 | 63.86 ± 0.51 | 62.63 ± 1.15 | 89.27 ± 0.05 |
| | NCNC | 29.01 ± 3.83 | 64.03 ± 3.67 | 25.70 ± 4.48 | 65.97 ± 1.03 | 62.61 ± 0.76 | 89.82 ± 0.43 |
| | LPFormer | 39.42 ± 5.78 | 65.42 ± 4.65 | 40.17 ± 1.92 | 68.14 ± 0.51 | 63.32 ± 0.63 | 89.81 ± 0.13 |
| ● | MPLP+ | N/A | N/A | N/A | 66.99 ± 0.40 | 65.24 ± 1.50 | 90.72 ± 0.12 |
| | Refined-GAE | N/A | N/A | N/A | 66.11 ± 0.35 | 78.41 ± 0.83 | 88.74 ± 0.06 |
| ● | GCN | 32.50 ± 6.87 | 50.01 ± 6.04 | 19.94 ± 4.24 | 54.96 ± 3.18 | 29.57 ± 2.90 | 84.85 ± 0.07 |
| | SAGE | 37.83 ± 7.75 | 47.84 ± 6.39 | 22.74 ± 5.47 | 59.44 ± 1.37 | 41.02 ± 1.94 | 83.06 ± 0.09 |
| | NBFNet | 37.69 ± 3.97 | 38.17 ± 3.06 | 44.73 ± 2.12 | N/A | N/A | N/A |
| | PENCIL w/o Features (Ours) | 42.23 ± 1.98 | 47.51 ± 3.09 | 38.28 ± 2.59 | 66.88 ± 0.34 | 73.85 ± 0.40 | 86.74 ± 0.26 |
| | PENCIL (Ours) | 32.12 ± 3.04 | 43.74 ± 5.47 | 38.34 ± 5.14 | 66.56 ± 0.19 | 79.54 ± 0.07 | 86.86 ± 0.20 |

*Table 2.* Results under HeaRT. N/A means results are not available. Colored results indicate the 1st, 2nd, and 3rd-best performance in MRR. Category markers: ● Heuristic-informed, ● ID-based, and ● Heuristic-agnostic + ID-free.

| Type | Model | cora | citeseer | pubmed | ogbl-collab | ogbl-ppa | ogbl-ddi | ogbl-citation2 |
|---|---|---|---|---|---|---|---|---|
| ● | SEAL | 10.67 ± 3.46 | 13.16 ± 1.66 | 5.88 ± 0.53 | 6.43 ± 0.32 | 29.71 ± 0.71 | 9.99 ± 0.90 | 20.60 ± 1.28 |
| | Neo-GNN | 13.95 ± 0.39 | 17.34 ± 0.84 | 7.74 ± 0.30 | 5.23 ± 0.90 | 21.68 ± 1.14 | 10.86 ± 2.16 | 16.12 ± 0.25 |
| | BUDDY | 13.71 ± 0.59 | 22.84 ± 0.36 | 7.56 ± 0.18 | 5.67 ± 0.36 | 27.70 ± 0.33 | 12.43 ± 0.50 | 19.17 ± 0.20 |
| | NCN | 14.66 ± 0.95 | 28.65 ± 1.21 | 5.84 ± 0.22 | 5.09 ± 0.38 | 35.06 ± 0.26 | 12.86 ± 0.78 | 23.35 ± 0.28 |
| | NCNC | 14.98 ± 1.00 | 24.10 ± 0.65 | 8.58 ± 0.59 | 4.73 ± 0.86 | 33.52 ± 0.26 | N/A | 19.61 ± 0.54 |
| | LPFormer | 16.80 ± 0.52 | 26.34 ± 0.67 | 9.99 ± 0.52 | 7.62 ± 0.26 | 40.25 ± 0.24 | 13.20 ± 0.54 | 24.70 ± 0.55 |
| ● | MPLP+ | N/A | N/A | N/A | 6.79 | 41.40 | N/A | 23.11 |
| ● | GCN | 16.61 ± 0.30 | 21.09 ± 0.88 | 7.13 ± 0.27 | 6.09 ± 0.38 | 26.94 ± 0.48 | 13.46 ± 0.34 | 19.98 ± 0.35 |
| | SAGE | 14.74 ± 0.69 | 21.09 ± 1.15 | 9.40 ± 0.70 | 5.53 ± 0.50 | 27.27 ± 0.30 | 12.60 ± 0.72 | 22.05 ± 0.12 |
| | NBFNet | 13.56 ± 0.58 | 14.29 ± 0.80 | N/A | N/A | N/A | N/A | N/A |
| | PENCIL w/o Features (Ours) | 14.63 ± 0.52 | 16.50 ± 0.31 | 7.05 ± 0.17 | 5.25 ± 0.01 | 44.57 ± 0.15 | 14.07 ± 0.24 | 23.36 ± 0.07 |
| | PENCIL (Ours) | 13.13 ± 0.53 | 16.80 ± 1.43 | 8.88 ± 0.49 | 5.40 ± 0.05 | 45.43 ± 0.31 | N/A | 23.43 ± 0.14 |

additional layers are introduced.

## 6. Related Work

**Link Prediction with GNNs.** A long line of link predictors augment a base MPNN with auxiliary structural signals. These include overlap-driven/local structure modules (e.g., NCN, Neo-GNN, BUDDY) (Wang et al., 2024; Yun et al., 2021; Chamberlain et al., 2023), explicit pairwise encodings (e.g., Distance Encoding, DRNL) (Li et al., 2020; Zhang & Chen, 2018), and global graph heuristics or per-node ID embeddings (e.g., LPFormer, MPLP, Refined-GAE) (Shomer et al., 2024; Dong et al., 2024; Ma et al., 2025b). This stands in contrast to PENCIL, which solely relies on random vector embeddings as node representations (see Section 3). Although PENCIL is formulated as a purely subgraph-based predictor, it could be extended in a scalable hybrid fashion by applying subgraph-based scoring to candidates within the extracted subgraph while using a two-tower retriever for candidates outside it, similarly to ContextGNN (Yuan et al., 2025). We leave the investigation of more scalable

approaches for future work.

**Expressive Powers.** The expressive power of standard GNNs is upper-bounded by the 1-WL test (Xu et al., 2019). By contrast, Transformers with Laplacian positional encodings are universal approximators and can exceed standard GNN expressivity (Kreuzer et al., 2021). Although PEN-CIL uses no such PEs, it fits within established expressivity frameworks via relational pooling (Murphy et al., 2019; Chen et al., 2020; Zhou et al., 2023), which symmetrizes a base predictor by averaging over input permutations. Under this lens, PENCIL's randomized vector assignments act as lightweight symmetry breaking, enabling a principled comparison to other link predictors (Lachi et al., 2026). More broadly, random features can already be highly expressive (Sato et al., 2021; Abboud et al., 2021); to our knowledge, PENCIL is the first to formalize link prediction as a randomized Transformer-based predictor and to show that permutation invariance is attainable in distribution.

**Graph Transformers and PEs/SEs.** Prior work on graph Transformers spans a range of choices for injecting struc-

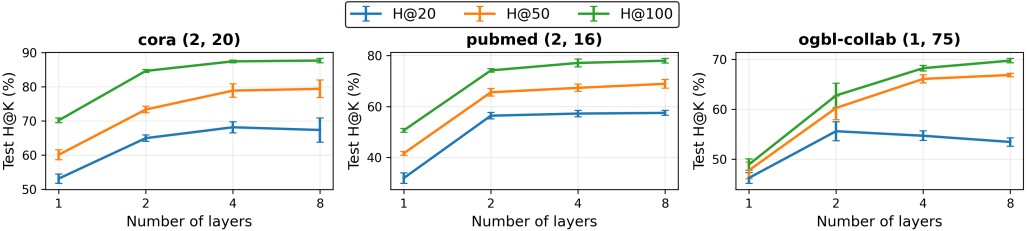

*Figure 5.* Effect of model depth on different performance metrics across the `cora`, `pubmed`, and `ogbl-collab` datasets.

tural bias. To name a few, Graph Transformer (Dwivedi & Bresson, 2021) uses Laplacian eigenvectors as PE, Graph-Trans (Jain et al., 2021) stacks a vanilla Transformer on top of GNN outputs, Graphormer (Ying et al., 2021) injects multiple structural biases into full-graph attention, and NeuralWalker (Chen et al., 2025) linearizes graphs via random walks. More broadly, because graphs admit no canonical sequence representation, prior work tokenizes structure in various ways—including hops (Chen et al., 2023), euler-ization paths (Zhao et al., 2025a), edge lists (Kim et al., 2022; Sanford et al., 2024), or nodes (Dwivedi et al., 2023; Yehudai et al., 2026; Jain et al., 2021). Most of the above graph-Transformer models are developed graph-level (or node-level) prediction, whereas link prediction poses a distinct dyadic setting with different supervision and deployment constraints. To our knowledge, PENCIL is the first to demonstrate that a plain Transformer can serve as a highly effective link predictor under these constraints. By contrast, although LPFormer employs attention for link prediction (Shomer et al., 2024), it relies on a PPR-based PE and therefore falls outside our deployment setting.

## 7. Conclusion, Limitations and Future Work

PENCIL is not simply that a Transformer can be competitive for link prediction, but that we provide, the first systematic explanation of what enables a vanilla bidirectional Transformer to do so without the strong PEs typically required by GTs. PENCIL bridges the gap between high-expressivity link prediction and realistic deployment by utilizing a vanilla Transformer over local subgraphs, entirely bypassing the need for structural encodings or per-node ID dependencies. Our results demonstrate that this architecture extracts richer structural signals than standard GNNs, achieving state-of-the-art performance with significantly fewer parameters and proving that simple, hardware-efficient designs are sufficient for large-scale link prediction. We believe that PENCIL is a principled step toward establishing plain Transformers as a legitimate and theoretically grounded paradigm for link prediction, with broader implications for graph machine learning. Finally, sampling budget serves as a natural knob controlling the expressive power accessible in practice. More broadly, this suggests that advances in hardware

and systems for processing larger sampled subgraphs can directly expand the practical capabilities of models like PENCIL.

PENCIL requires a context subgraph for each candidate link; consequently, its GPU compute and memory scale linearly with the number of candidate links. Developing resource-efficient variants—e.g., caching subgraph computations or hybridizing with retrieval-style scoring—remains future work. Moreover, PENCIL's empirical gains are most pronounced on large-scale datasets (Section 5), suggesting that reducing its data requirements through pretraining is a promising avenue. Finally, while our expressivity analysis shows that PENCIL can degenerate to SEAL, a tighter theoretical characterization of full-attention Transformer link predictors is still lacking and would enable clearer comparisons across emerging Transformer-based approaches.

## Acknowledgements

Quang Truong, Yu Song, and Jiliang Tang are supported by the National Science Foundation (NSF) under grant numbers CNS2321416, IIS2212032, IIS2212144, IIS 2504089, DUE2234015, CNS2246050, DRL2405483 and IOS2035472, the Michigan Department of Agriculture and Rural Development, US Dept of Commerce, Gates Foundation, Amazon Faculty Award, Meta, NVIDIA, Microsoft and SNAP.

## Impact Statement

This work advances graph machine learning for link prediction by introducing PENCIL, a deployment-oriented Transformer-based link predictor that operates on fixed-budget sampled subgraphs. Potential applications include recommendation, knowledge graph completion, and biological interaction discovery. As with any link prediction system, PENCIL may be used in settings that affect individuals (e.g., recommender systems or social inference), where erroneous, biased, or harmful links could be suggested or amplified. Accordingly, we encourage domain-specific evaluation and responsible deployment practices in high-stakes settings.

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

# A. Adjacency Reconstruction for Multiplicative Residual Connection

The node-adjacency encoding scheme (Yehudai et al., 2026) used in PENCIL is constructed so that the sampled subgraph connectivity can be recovered directly from the input tensor $\tilde{\mathbf{X}}$, eliminating the need to separately batch an adjacency structure.

Recall that each mini-batch contains $B$ sampled subgraphs. For sample $b$, let $N_b$ be the number of sampled *context* nodes, and define $N_B = \max_{b \in [B]} N_b$ as the maximum number of context nodes in the batch after padding. Let $N_{\max}$ be the global sampling budget (maximum allowed context nodes per subgraph over the dataset). The tokenized input $\tilde{\mathbf{X}} \in \mathbb{R}^{B \times (N_B+2) \times (2N_{\max}+2)}$ contains $N_B$ context tokens plus two *task* tokens (for $v_{\text{src}}$ and $v_{\text{dst}}$). Each token row concatenates (i) a padded one-hot identifier over the $N_{\max}$ context-index slots, (ii) a padded adjacency-indicator row over the same $N_{\max}$ slots, and (iii) a 2-bit role flag (see Figure 2). Importantly, the two task-token rows copy the identifier and adjacency parts of their corresponding endpoint context tokens, differing only in the role flag.

**Reconstructing a context-sourced adjacency operator.**  We recover the subgraph adjacency operator by slicing out the identifier and adjacency blocks from $\tilde{\mathbf{X}}$. Using Python-style indexing, define

$$\tilde{\mathbf{X}}^{\text{id}} = \tilde{\mathbf{X}}_{:,:,\,0:N_B}, \qquad \tilde{\mathbf{X}}^{\text{adj}} = \tilde{\mathbf{X}}_{:,:,\,N_{\max}:N_{\max}+N_B}. \tag{6}$$

Both slices have shape $B \times (N_B + 2) \times N_B$. We slice only the first $N_B$ columns (rather than all $N_{\max}$) to restrict the operator to the active, non-padded context slots in the current batch.

Intuitively, $\tilde{\mathbf{X}}^{\text{adj}}$ provides the *off-diagonal* neighborhood indicators: the entry $\tilde{\mathbf{X}}^{\text{adj}}_{b,i,j} = 1$ indicates that, in sample $b$, token $i$ is connected to the $j$-th context token (under the subgraph indexing used to form $\tilde{\mathbf{X}}$). The slice $\tilde{\mathbf{X}}^{\text{id}}$ provides an *identity link* to the corresponding context index, i.e., it plays the role of adding self-loops for context tokens, and (because task tokens copy the endpoint identifier) it also ensures that each task token can directly aggregate its endpoint's own context representation through the propagation branch.

We therefore define a *context-sourced* adjacency operator (rows are all tokens; columns are context tokens only):

$$\tilde{\mathbf{A}}_{\text{src}} = \tilde{\mathbf{X}}^{\text{adj}} + \tilde{\mathbf{X}}^{\text{id}} \in \mathbb{R}^{B \times (N_B+2) \times N_B}. \tag{7}$$

The name $\tilde{\mathbf{A}}_{\text{src}}$ emphasizes that only the $N_B$ context tokens act as *message sources* in this operator, while *all* $N_B + 2$ tokens (including the two task tokens) can act as *receivers*.

**Extending to a square operator.**  For the multiplicative residual branch, it is convenient to work with a square $(N_B + 2) \times (N_B + 2)$ operator. Since the encoding does not allocate dedicated column slots for the two task tokens (task tokens are virtual readout tokens rather than sampled subgraph nodes), we append two all-zero columns:

$$\tilde{\mathbf{A}} = \begin{bmatrix} \tilde{\mathbf{A}}_{\text{src}} & \mathbf{0} \end{bmatrix}, \qquad \mathbf{0} \in \mathbb{R}^{B \times (N_B+2) \times 2}. \tag{8}$$

This yields $\tilde{\mathbf{A}} \in \mathbb{R}^{B \times (N_B+2) \times (N_B+2)}$. By construction, the task tokens have zero outgoing columns and therefore do not act as sources in the propagation step; they behave as *receive-only (sink) tokens*, aggregating messages from the context subgraph. This design is intentional: the Transformer self-attention already enables dense interactions among all tokens within each block, while the propagation branch injects an explicit one-hop, structure-respecting aggregation from the sampled context graph into both context and task representations. Empirically, we adopt the row-normalized form $\mathbf{D}^{-1}\tilde{\mathbf{A}}$ to enhance training stability.

# B. Supplementary Materials for Theoretical Analysis

All theoretical results are restated first before the proof for the reader's convenience.

## B.1. Proof for Theorem 4.1

We first state the formal statement of Theorem 4.1.

**Theorem B.1.** *Let $\mathcal{V} = \{0, 1, \ldots, N-1\}$ and let $\mathbf{A} \in \{0,1\}^{N \times N}$ be the adjacency matrix of a graph on $\mathcal{V}$. Fix an ordered query pair $(u, v) \in \mathcal{V} \times \mathcal{V}$ with $u \neq v$. Define the endpoint-constrained permutation set*

$$\Gamma_{u,v} := \{\rho \in S_N : \rho(u) = 0, \ \rho(v) = 1\}.$$

*Let $\rho \sim \text{Unif}(\Gamma_{u,v})$ be a random permutation drawn uniformly from the endpoint-constrained permutation set and let $f$ be any deterministic measurable function. Define the randomized predictor*

$$S(\mathbf{A}; u, v) := f\left(\mathbf{P}_\rho \, \mathbf{A} \, \mathbf{P}_\rho^\top\right).$$

*Then for any permutation $\pi \in S_N$, which is the symmetric group on $N$ elements, letting*

$$\mathbf{A}' := \mathbf{P}_\pi \, \mathbf{A} \, \mathbf{P}_\pi^\top, \qquad u' := \pi(u), \qquad v' := \pi(v).$$

*Then, the distribution of $S(\mathbf{A}; u, v)$ is identical to that of $S(\mathbf{A}'; u', v')$:*

$$S(\mathbf{A}; u, v) \stackrel{d}{=} S(\mathbf{A}'; u', v').$$

The endpoint-constrained randomness $\rho$ models the stochastic reindexing of non-query nodes (Section 2). In contrast, $\pi \in S_N$ denotes an arbitrary node relabeling, reflecting the standard permutation symmetry that graph predictors should satisfy. Theorem B.1 establishes that $S$ is permutation-invariant in distribution under these relabelings. Next, we proceed with the proof.

*Proof.* Fix any $\pi \in S_N$. Let

$$\mathbf{A}' := \mathbf{P}_\pi \mathbf{A} \mathbf{P}_\pi^\top, \qquad u' := \pi(u), \qquad v' := \pi(v),$$

and let $\rho' \sim \text{Unif}(\Gamma_{u',v'})$. By definition,

$$S(\mathbf{A}'; u', v') = f\left(\mathbf{P}_{\rho'} \mathbf{A}' \mathbf{P}_{\rho'}^\top\right).$$

To prove $S(\mathbf{A}; u, v) \stackrel{d}{=} S(\mathbf{A}'; u', v')$, it suffices to show that for every measurable set $B \subseteq \mathbb{R}$,

$$\Pr\left(S(\mathbf{A}; u, v) \in B\right) = \Pr\left(S(\mathbf{A}'; u', v') \in B\right).$$

Since $\rho'$ is uniform over the finite set $\Gamma_{u',v'}$, we can expand

$$\Pr\left(S(\mathbf{A}'; u', v') \in B\right) = \Pr\left(f\left(\mathbf{P}_{\rho'} \mathbf{A}' \mathbf{P}_{\rho'}^\top\right) \in B\right) = \frac{1}{|\Gamma_{u',v'}|} \sum_{\rho' \in \Gamma_{u',v'}} \mathbb{1}\left\{f\left(\mathbf{P}_{\rho'} \mathbf{A}' \mathbf{P}_{\rho'}^\top\right) \in B\right\}. \tag{9}$$

Substituting $\mathbf{A}' = \mathbf{P}_\pi \mathbf{A} \mathbf{P}_\pi^\top$ yields

$$\mathbf{P}_{\rho'} \mathbf{A}' \mathbf{P}_{\rho'}^\top = \mathbf{P}_{\rho'}(\mathbf{P}_\pi \mathbf{A} \mathbf{P}_\pi^\top)\mathbf{P}_{\rho'}^\top = (\mathbf{P}_{\rho'}\mathbf{P}_\pi)\,\mathbf{A}\,(\mathbf{P}_{\rho'}\mathbf{P}_\pi)^\top = \mathbf{P}_{\rho' \circ \pi}\,\mathbf{A}\,\mathbf{P}_{\rho' \circ \pi}^\top. \tag{10}$$

Plugging (10) into (9) gives

$$\Pr\left(S(\mathbf{A}'; u', v') \in B\right) = \frac{1}{|\Gamma_{u',v'}|} \sum_{\rho' \in \Gamma_{u',v'}} \mathbb{1}\left\{f\left(\mathbf{P}_{\rho' \circ \pi}\,\mathbf{A}\,\mathbf{P}_{\rho' \circ \pi}^\top\right) \in B\right\}. \tag{11}$$

Let $\rho := \rho' \circ \pi$. We claim that the map $\varphi_\pi : \Gamma_{u',v'} \to \Gamma_{u,v}$ such that $\varphi_\pi(\rho') = \rho' \circ \pi$ is a bijection. Indeed, if $\rho' \in \Gamma_{u',v'}$, we have:

$$(\rho' \circ \pi)(u) = \rho'(\pi(u)) = \rho'(u') = 0, \qquad (\rho' \circ \pi)(v) = \rho'(\pi(v)) = \rho'(v') = 1,$$

so $\rho' \circ \pi \in \Gamma_{u,v}$. Conversely, if $\rho \in \Gamma_{u,v}$ then $\rho \circ \pi^{-1} \in \Gamma_{u',v'}$ and $\varphi_\pi(\rho \circ \pi^{-1}) = \rho$. Hence, $\varphi_\pi$ is bijective, and in particular $|\Gamma_{u',v'}| = |\Gamma_{u,v}|$.

Therefore, we can rewrite the sum in (11) over $\rho \in \Gamma_{u,v}$ to obtain

$$\Pr\left(S(\mathbf{A}'; u', v') \in B\right) = \frac{1}{|\Gamma_{u,v}|} \sum_{\rho \in \Gamma_{u,v}} \mathbb{1}\left\{f\left(\mathbf{P}_\rho \, \mathbf{A} \, \mathbf{P}_\rho^\top\right) \in B\right\}$$

$$= \Pr\left(f\left(\mathbf{P}_\rho \mathbf{A} \mathbf{P}_\rho^\top\right) \in B\right)$$

$$= \Pr\left(S(\mathbf{A}; u, v) \in B\right), \tag{12}$$

where the second equality uses $\rho \sim \text{Unif}(\Gamma_{u,v})$. (12) holds for all measurable $B$, which concludes the proof. $\square$

## B.2. Neural Bellman–Ford Networks

We first summarize the generalized Bellman–Ford algorithm (Baras & Theodorakopoulos, 2010), rewriting the notation for consistency with our presentation. For the formulation used in knowledge graph setting and its connection to NBFNet, see (Zhu et al., 2021).

**Definition B.2** (Generalized Bellman–Ford Algorithm). Let $G = (\mathcal{V}, \mathcal{E})$ be a (directed) graph and fix a source node $u$. Let $(\mathcal{S}, \oplus, \otimes)$ be a semiring with additive identity $\circledcirc$ and multiplicative identity $\circledone$. Define the incoming edge set of $v$ as $\mathcal{I}(v) := \{ (x, v) \in \mathcal{E} \}$. Let $\mathbf{h}_v^{(t)} \in \mathcal{S}$ denote the node representation of $v$ at iteration $t$, and let $\mathbf{w}(x, v) \in \mathcal{S}$ denote the edge representation of edge $(x, v)$. Here $\mathbb{1}(\cdot)$ is an indicator that outputs $\circledone$ if $v = u$ and $\circledcirc$ otherwise.

The generalized Bellman–Ford updates are

$$\mathbf{h}_v^{(0)} \leftarrow \mathbb{1}(v = u), \tag{13}$$

$$\mathbf{h}_v^{(t)} \leftarrow \left( \bigoplus_{(x,v) \in \mathcal{I}(v)} \mathbf{h}_x^{(t-1)} \otimes \mathbf{w}(x, v) \right) \oplus \mathbf{h}_v^{(0)}. \tag{14}$$

It has been shown that, with appropriate choices of the semiring operators $(\oplus, \otimes)$, the generalized Bellman–Ford recursion recovers a broad class of path-based heuristics for link prediction (Zhu et al., 2021). NBFNet departs from the strict semiring setting by parameterizing the update with learnable functions: the indicator initialization is replaced by a learnable source embedding, $\otimes$ is instantiated as a message function, and $\oplus$ is instantiated as a permutation-invariant aggregation function. This modification yields a source-conditioned MPNN, where representations $\mathbf{h}_v^{(t)}$ are computed conditioned on the chosen source node $u$ (You et al., 2021).

### B.3. Proof of Proposition 4.2

**Proposition 4.2.** *There exists a parameter setting of PENCIL under which its layerwise update reduces to a source-conditioned MPNN, with readout at the canonical destination token $v_1$.*

*Proof.* By canonicalization, the source is always token $v_0$, so we can choose the input projection s.t. $\mathbf{h}_v^{(0)} = \mathbb{1}(v = v_0)\mathbf{e}$ for a learnable $\mathbf{e} \in \mathbb{R}^d$, matching NBFNet's source-conditioned initialization. Setting $\mathbf{T}_k$ to map tokens to zeroes for all $k$ removes the attention branch, and Eq. 3 reduces to a propagation on $\tilde{\mathbf{A}}$. Reading out at the canonical destination token $v_1$ yields a source-conditioned link score. $\square$

### B.4. Proof of Corollary 4.3

**Corollary 4.3.** *Under suitable parameter settings and operator choices, PENCIL can realize a broad class of classical path-based link prediction scores and graph algorithms, including Katz index, Personalized PageRank, SPD, widest path, and most reliable path.*

*Proof.* By Proposition 4.2, there exists a parameter setting under which PENCIL reduces to an NBFNet-style propagation model. (Zhu et al., 2021) shows that the generalized Bellman–Ford recovers the listed heuristics under appropriate choices of the semiring operators and edge weights. $\square$

### B.5. Formal statements of Remark 4.4

The following results are established in (Dong et al., 2024); we restate them here with minor rephrasing to align with our notation and paper context.

**Theorem B.3.** *(Dong et al., 2024) Let $\mathbf{h}_v^{(0)} \in \mathbb{R}^d$ be initial node representations where each node vector is a zero-mean unit-norm vector in expectation. Under a single layer of sum-aggregation message passing, the inner product of any two node embeddings is an unbiased estimator of their common neighbor count:*

$$\mathbb{E}\left[ \mathbf{h}_u^{(1)} \cdot \mathbf{h}_v^{(1)} \right] = |\mathcal{N}(u) \cap \mathcal{N}(v)|. \tag{15}$$

*Table 3.* Model formulations expressed within the $k_\phi$-$k_\rho$-$m$ framework. A '/' indicates that the corresponding component is not included in the model. Table is extracted from (Lachi et al., 2026), where PENCIL is our contribution.

| Model | COMB | $g$ | $k_\phi$ | AGG | $\psi$ | $k_\rho$ | $m$ | $h$ |
|---|---|---|---|---|---|---|---|---|
| Pure GNN | / | $\odot$ | 1-WL | / | / | / | / | / |
| NCN | $\|$ | $\odot$ | 1-WL | $\sum$ | $\rho(i; G, \mathbf{X}^0)$ | 1-WL | 1 | $\mathbf{X}^0$ |
| ELPH | $\|$ | $\|$ | 1-WL | $\sum$ | $\rho(i; G, \mathbf{X}^1) \cdot \prod_{r=1}^m \prod_{d=1}^m \mathbb{1}_{dr}(i)$ | 1-WL | m | $\mathbf{x}_i^1 = 1$ |
| Neo-GNN | $\|$ | $\|$ | 1-WL | $\sum$ | $b \cdot \rho(i; G, \mathbf{X}^0)$ with $b = \sum_{r=1}^m \sum_{d=1}^m (\mathbf{A}^r)_{uv} \cdot (\mathbf{A}^d)_{uv}$ | 1-WL | m | $\mathbf{X}^0$ |
| SEAL | / | / | / | $\sum$ | $\rho(i; G, \mathbf{X}^D)$ | $1 - \|N^m(u,v)\|$-WL | m | $\mathbf{x}_i^D = \mathbf{x}_i^0 \| \min_{u,v}(\delta(i,u), \delta(i,v)) + 1$ |
| PENCIL | / | / | / | $\sum$ | $\psi_{\text{end}}(\rho(i; G, \tilde{\mathbf{X}}), u, v)$ with $\psi_{\text{end}}(z_i, u, v) = [\mathbb{1}(i=u) z_i \| \mathbb{1}(i=v) z_i]$ | $1 - \|N^m(u,v)\|$-WL | m | $\tilde{\mathbf{x}}_i = \mathbf{r}_i$ |

**Theorem B.4.** *(Dong et al., 2024) Under the same conditions as Theorem B.3, given $p$ and $q$ iterations of sum-aggregation message passing, the expected inner product between the resulting node embeddings satisfies*

$$\mathbb{E}\left[\mathbf{h}_u^{(p)} \cdot \mathbf{h}_v^{(q)}\right] = \sum_{k \in V} \left|\text{walks}^{(p)}(k, u)\right| \left|\text{walks}^{(q)}(k, v)\right|, \tag{16}$$

*where $\left|\text{walks}^{(\ell)}(a, b)\right|$ denotes the number of length-$\ell$ walks between nodes $a$ and $b$.*

### B.6. Proof of Proposition 4.5

**Proposition 4.5.** *There exists a parameter setting of PENCIL under which it can estimate heuristics stated in Theorem B.3 and Theorem B.4.*

*Proof.* We can choose the input projection such that, for each sampled token $v_i$, $i \in [N]$, $\mathbf{h}_{v_i}^{(0)} = \mathbf{e}_i$, where $\mathbf{e}_i \in \mathbb{R}^d$ is learnable. This is effectively equivalent to assigning a learnable random vector to each sampled node. Importantly, $\{\mathbf{e}_i\}_{i=1}^N$ can be chosen to be mutually orthonormal for $d \geq N_{\max}$. Setting $\mathbf{T}_k$ to map tokens to zeroes for all $k$ removes the attention branch, and Eq. 3 reduces to a sum-aggregation MPNN. The resulting model meets the conditions of Theorem B.3 and Theorem B.4, which concludes the proof. $\square$

### B.7. Proof of Proposition 4.8

**Proposition 4.8.** *For a fixed dimension $d \geq 2$, the Welch bound $W(N, d)$ is strictly increasing as a function of the set size $N$ for all $N > d$.*

*Proof.* Fix $d \geq 2$ and let $M > N > d$. Since $\sqrt{\cdot}$ is strictly increasing on $(0, \infty)$, it suffices to show

$$\frac{M - d}{M - 1} > \frac{N - d}{N - 1}.$$

Because $(M - 1)(N - 1) > 0$, cross-multiplying yields

$$\frac{M - d}{M - 1} > \frac{N - d}{N - 1} \iff (M - d)(N - 1) > (N - d)(M - 1).$$

Rearranging,

$$(M - d)(N - 1) - (N - d)(M - 1) = (M - N)(d - 1) > 0,$$

since $M > N$ and $d \geq 2$. Therefore $W(M, d) > W(N, d)$, and $W(N, d)$ is strictly increasing in $N$ for all $N > d$. $\square$

### B.8. Expressive Power of PENCIL

We first briefly review the $k_\phi$-$k_\rho$-$m$ framework proposed in (Lachi et al., 2026), which is the framework where our expressive power analysis is conducted. Then, we align PENCIL with the framework.

**Definition B.5** ($k_\phi$-$k_\rho$-$m$ framework). Let $G = (\mathcal{V}, \mathcal{E})$ be a graph with initial node-feature matrix $\mathbf{X}^0$. For brevity, we suppress the explicit dependence of all functions on $G$ unless needed.

A link-representation MPNN $M$ is said to belong to the $k_\phi$-$k_\rho$-$m$ framework if its output for a node pair $(u, v)$ can be written as

$$F\big((u, v), \mathbf{X}^0\big) = \text{COMB}\Big(g\big(\phi(u, \mathbf{X}^0), \phi(v, \mathbf{X}^0)\big), \text{AGG}\big(\{\, f(i, u, v, \mathbf{X}^0) \mid i \in \bigcup_{j=0}^{m} \mathcal{N}^j(u, v) \,\}\big)\Big), \quad (17)$$

$$f(i, u, v, \mathbf{X}^0) = \psi\big(\rho(i, h(u, v, \mathbf{X}^0)), u, v\big), \quad (18)$$

where:

- $\phi$ and $\rho$ are MPNNs with expressive power $k_\phi$ and $k_\rho$, respectively.

- For $m \geq 1$, $\mathcal{N}^m(v)$ denotes the set of nodes at exactly $m$ hops from $v$. For a pair $(u, v)$, define the joint $m$-hop neighborhood $\mathcal{N}^m(u, v) := \mathcal{N}^m(u) \cup \mathcal{N}^m(v)$.

- $h(u, v, \mathbf{X}^0) \in \mathbb{R}^{|\mathcal{V}| \times d}$ is a derived node-feature matrix computed from $\mathbf{X}^0$, optionally augmented with pair-specific information for $(u, v)$.

- $\psi$ applies a pair-dependent rescaling to message-passing representations, potentially using information specific to $(u, v)$.

- $g$ aggregates the node representations of endpoints $\phi(u, \mathbf{X}^0)$ and $\phi(v, \mathbf{X}^0)$ into an endpoints' representation.

- AGG is a permutation-invariant aggregation operator over the set of node representations in the selected neighborhood, and COMB merges the endpoints' representation and neighborhood representations.

Table 3 aligns existing link predictors—ELPH (Chamberlain et al., 2023), Neo-GNN (Yun et al., 2021), NCN (Wang et al., 2024), and SEAL (Zhang & Chen, 2018)—with the $k_\phi$-$k_\rho$-$m$ framework (Lachi et al., 2026). Our primary comparison is between PENCIL and SEAL; the remaining models are included for context (see (Lachi et al., 2026) for details).

PENCIL and SEAL fit the framework in a similar *link-centric* manner: neither model uses a separate endpoint encoder $\phi$ nor an explicit endpoint combiner $(g, \text{COMB})$. Instead, both compute node representations on the extracted enclosing subgraph and then aggregate them into a link representation. Consequently, the relevant components for distinguishing the two models are AGG, $\psi$, $k_\rho$, $m$, and $h$.

Both methods operate on the $m$-hop enclosing neighborhood of $(u, v)$, so they share the same $m$ in Table 3. SEAL constructs a distance-aware feature matrix $\mathbf{X}^D$ using DRNL labels (Zhang & Chen, 2018) and applies an MPNN $\rho$ on this subgraph. In the framework, this corresponds to $h(u, v, \mathbf{X}^0) = \mathbf{X}^D$ and a pair-independent identity mapping $\psi(z_i, u, v) = z_i$, so the neighborhood-level representation reduces to aggregation over all nodes in the $m$-hop enclosing neighborhood: $\sum_{i \in \bigcup_{j=0}^{m} \mathcal{N}^j(u, v)} \rho(i; G, \mathbf{X}^D)$.

In contrast, PENCIL uses an index-dependent input embedding $\tilde{\mathbf{X}}$ for the sampled subgraph nodes (Section 3), which in Table 3 is captured by $h(u, v, \mathbf{X}^0) = \tilde{\mathbf{X}}$ with $\tilde{\mathbf{x}}_i = \mathbf{r}_i$, where $\mathbf{r}_i$ is a random vector induced by the input projection matrix. The final link representation is formed by extracting and concatenating the endpoint-specific components, which can be written as an $\text{AGG} = \sum$ over a gated construction $\psi_{\text{end}}(z_i, u, v) = [\mathbb{1}(i = u)\, z_i \parallel \mathbb{1}(i = v)\, z_i]$. This choice of $\psi$ is a convenient readout that isolates the endpoint representations. Next, we restate Theorem 4.9 and provide its proof.

**Theorem 4.9.** *Under the $k_\phi$-$k_\rho$-$m$ framework, for suitable parameter settings, PENCIL with LRP is not less expressive than SEAL under the same sampling constraint.*

*Proof.* We first note that both models are *local* around the queried edge: for a fixed integer $m$, the prediction for $(u, v)$ is a function only of the $m$-hop enclosing neighborhood $N^m(u, v)$. Hence, the two models share the same locality parameter $m$ in the $k_\phi$-$k_\rho$-$m$ framework.

Next, consider PENCIL under a parameter setting with $\mathbf{T}_k = \mathrm{Id}$ for all $k$ and an input projection such that $\mathbf{h}_v^{(0)} = \mathbb{1}(v = v_i)\mathbf{e}_i$ for learnable vectors $\mathbf{e}_i \in \mathbb{R}^d$. Under this setting, PENCIL reduces to a 1-WL MPNN operating on the sampled subgraph (see Eq. 3). Define the endpoint-constrained permutation set over the sampled subgraph as

$$\Gamma_{u,v} := \{\pi \in S_{|N^m(u,v)|} : \pi(u) = 0, \ \pi(v) = 1\},$$

where $S_N$ is the symmetric group on $N$ elements. With LRP, we aggregate the MPNN outputs over the $|\Gamma_{u,v}|$ endpoint-preserving relabelings; by (Zhou et al., 2023), this yields 1-$|N^m(u, v)|$-WL power, i.e., the same $k_\rho$ as SEAL on $N^m(u, v)$ (Lachi et al., 2026). Therefore, since two models have the same $m$ and $k_\rho$, the theorem follows according to Theorem 3.2 in (Lachi et al., 2026). $\square$

Under the expressivity ordering in (Lachi et al., 2026), the above 1-$|N^m(u, v)|$-WL guarantee is strictly stronger than the expressivity attributed to several heuristics-based predictors, including ELPH (Chamberlain et al., 2023), NCN (Wang et al., 2024), and Neo-GNN (Yun et al., 2021). Importantly, this result should *not* be interpreted as an upper bound on PENCIL: it only establishes that PENCIL can *degenerate* to 1-$|N^m(u, v)|$-WL expressivity under a particular parameter setting. Characterizing the expressive power of the full attention-enabled PENCIL remains an interesting direction for future work.

Complementarily, (Liang et al., 2024) argues that SEAL's Double Radius Node Labeling (DRNL) may not reliably expose common-neighbor signals in practice, since subsequent message passing and aggregation can wash out this information. In contrast, PENCIL's input projection injects token-specific vectors that make such neighborhood identity information directly accessible, as discussed and empirically validated in previous sections. Finally, we do not equip PENCIL with LRP in our experiments; empirically, we find that using a single labeled subgraph already suffices, as shown by experiments covered in Section 5 and Appendix C.5.

# C. Experiments

This section presents supplementary information regarding the experiments in Section 5, as well as further investigations into input projection matrix initialization and computational complexity.

Regarding practical implementation, we employ the ShaDowKHop sampler (Zeng et al., 2021), utilizing the implementation provided by GraphGPT (Zhao et al., 2025a). This sampler requires two hyperparameters: depth $d$ and the number of neighbors per node $n$. Where necessary to distinguish configurations, we append the suffix $(d, n)$ to dataset names to denote the specific parameters used. For the Transformer layer, we employ the BERT implementation (Devlin et al., 2019) provided by Hugging Face (Wolf et al., 2020). All codes are written in Pytorch (Paszke et al., 2019) and Pytorch Geometric (PyG) (Fey & Lenssen, 2019; Fey et al., 2025).

## C.1. Pairwise Heuristic Estimation

In this section, we describe the pairwise heuristics considered in our experiments, along with the normalization procedure applied to the raw heuristic scores so they can be used as regression targets, since many exhibit heavy-tailed distributions. A unified three-stage preprocessing protocol is used across all heuristics: (i) logarithmic transformation to reduce skewness; (ii) range mapping (normalization or standardization) to align scales; and (iii) robust clipping to prevent rare outliers from dominating the loss gradient. We mirror the link prediction pipeline for the heuristic regression task, including the sampling of one negative edge per positive training instance. However, we replace the binary classification target with a normalized heuristic score for regression. All experiments utilize the standard link prediction splits defined in (Li et al., 2023). We compare PENCIL against standard GNN baselines—GCN, GAT, and GraphSAGE (Kipf & Welling, 2017; Veličković et al., 2018; Hamilton et al., 2017)—as well as identity-aware variants that augment message passing with explicit markers for the query nodes (e.g., source/target indicators) (You et al., 2021; Zhu et al., 2021; Zhang et al., 2021). These approaches share the common principle of injecting node identity information to facilitate reasoning about a specific pair, and have been shown to scale to large relational settings (Robinson et al., 2024; Yuan et al., 2025). Since we do not use node features, both GNNs and PENCIL rely solely on graph structure: GNNs take all-ones node inputs and learn representations through message passing, whereas PENCIL uses the structural encoding from Section 2 and learns representations via self-attention over the encoded sequence. The results are shown in Figure 4 and 6. The hyperparameter configurations for GNNs and PENCIL in the pairwise heuristic estimation experiments are summarized in Table 4. The hidden size of PENCIL was

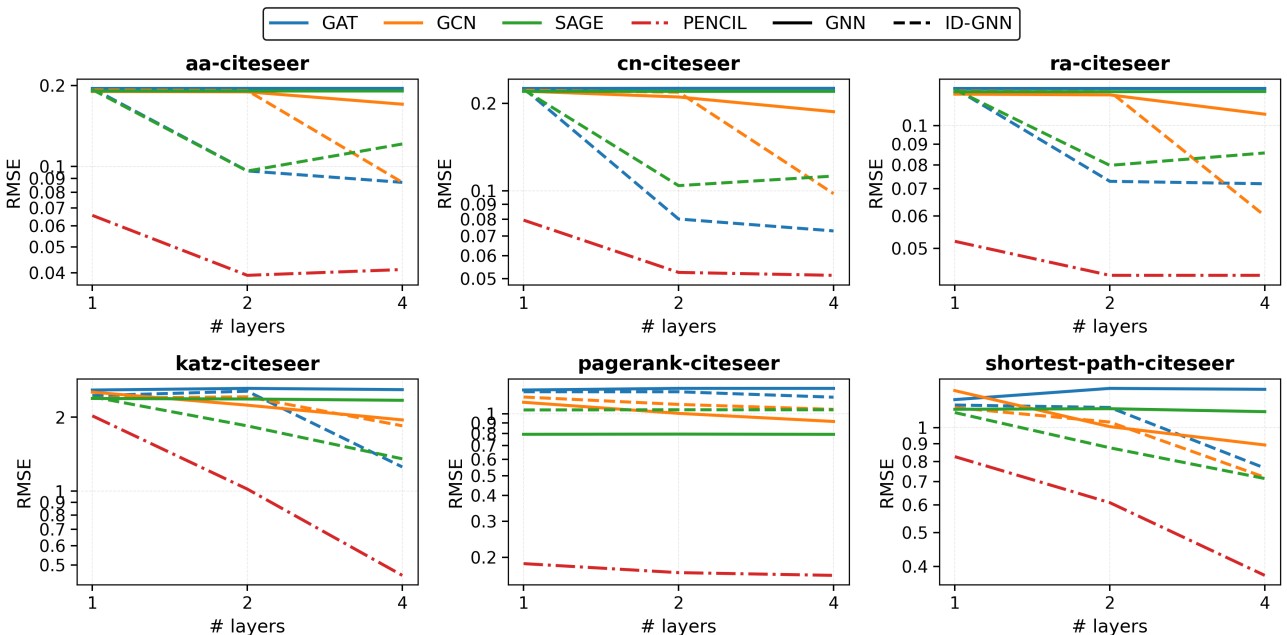

*Figure 6.* RMSE of PENCIL and other GNNs for estimating pairwise heuristics on the `citeseer` dataset.

*Table 4.* Hyperparameter configurations of GNNs and PENCIL for the pairwise heuristic estimation experiments. N/A means the hyperparameter is not applicable.

| Hyperparameter | GNNs | PENCIL |
|---|---|---|
| Sampling Configuration | (2, 20) | (2, 20) |
| Hidden Size | 512 | 256 |
| Intermediate Size | N/A | 512 |
| # Attention Heads | N/A | 4 |
| Effective Batch Size | 2048 | 2048 |
| Learning Rate | 2.0E-04 | 2.0E-04 |
| Weight Decay | 0.01 | 0.01 |

constrained to yield comparable parameter counts across all models. Models were trained for 200 epochs for CN, AA, RA, and PageRank, while the Katz index and SPD experiments required 700 epochs.

**Local Overlap Heuristics.** Three local overlap scores based on the shared one-hop neighborhood of a candidate pair $(u, v)$ are considered:

$$\text{CN}(u,v) := \big|\mathcal{N}(u) \cap \mathcal{N}(v)\big|, \tag{19}$$

$$\text{AA}(u,v) := \sum_{w \in \mathcal{N}(u) \cap \mathcal{N}(v)} \frac{1}{\log(\deg(w))}, \tag{20}$$

$$\text{RA}(u,v) := \sum_{w \in \mathcal{N}(u) \cap \mathcal{N}(v)} \frac{1}{\deg(w)}. \tag{21}$$

These heuristics quantify structural overlap between $u$ and $v$: Common Neighbors (CN) (Newman, 2001) counts the number of common neighbors, whereas Adamic–Adar (AA) (Adamic & Adar, 2003) and Resource Allocation (RA) (Zhou et al., 2009) discount common neighbors $w$ with large degree, thereby emphasizing rarer shared neighbors. Here, $\mathcal{N}(u)$ denotes the set of one-hop neighbors of node $u$, and $\deg(w) := |\mathcal{N}(w)|$ denotes the degree of node $w$. The summations range over the shared neighbors $w \in \mathcal{N}(u) \cap \mathcal{N}(v)$. Since these scores are non-negative and typically right-skewed, a log transformation $s \mapsto \log(1 + s)$ is applied, where $s \in \{\text{CN}(u,v), \text{AA}(u,v), \text{RA}(u,v)\}$. This is followed by min–max

*Table 5.* Hyperparameter configurations of PENCIL for the original benchmark settings.

| Hyperparameter | cora | citeseer | pubmed | ogbl-collab | ogbl-ppa | ogbl-citation2 |
|---|---|---|---|---|---|---|
| Sampling Configuration | (2, 20) | (2, 20) | (2, 16) | (1, 75) | (1, 150) | (1, 40) |
| Hidden Size | 512 | 1024 | 512 | 512 | 512 | 512 |
| Intermediate Size | 2048 | 2048 | 2048 | 2048 | 2048 | 2048 |
| # Layers | 8 | 2 | 8 | 8 | 4 | 3 |
| # Attention Heads | 8 | 8 | 8 | 8 | 8 | 8 |
| Effective Batch Size | 2048 | 2048 | 4096 | 8192 | 16384 | 65536 |
| Learning Rate | 1.0E-04 | 1.0E-04 | 1.0E-04 | 1.0E-04 | 1.0E-04 | 1.0E-04 |
| Weight Decay | 0.01 | 0.01 | 0.01 | 0.01 | 0.01 | 0.01 |
| # Epochs | 150 | 150 | 80 | 10 | 15 | 0.5 |

scaling that maps $[0, P_{99.99}]$ to $[0, 1]$, where $P_{99.99}$ denotes the 99.99$^{\text{th}}$ percentile of the log-transformed scores. Values exceeding this threshold are clipped to 1.

**Katz Index.** The Katz score for a pair $(u, v)$ is defined as a damped sum over walks of all lengths (Katz, 1953):

$$\text{Katz}_\beta(u, v) := \sum_{\ell=1}^{\infty} \beta^\ell \, (\mathbf{A}^\ell)_{uv}. \tag{22}$$

This global similarity measure aggregates multi-hop connectivity between $u$ and $v$ while exponentially down-weighting longer walks. Here, $\mathbf{A} \in \{0, 1\}^{|V| \times |V|}$ is the adjacency matrix, $(\mathbf{A}^\ell)_{uv}$ is the number of length-$\ell$ walks from $u$ to $v$, and $\beta \in (0, 1)$ is a damping factor (we use $\beta = 0.005$) that emphasizes shorter walks. Katz scores are transformed as $\log(k + \epsilon)$, where $k = \text{Katz}_\beta(u, v)$ and $\epsilon = 10^{-20}$, and then standardized to zero mean and unit variance. For robustness, the standardized scores are clipped in the upper tail at the 99$^{\text{th}}$ percentile.

**Shortest Path Distance.** The shortest-path distance between nodes $u$ and $v$ is defined as

$$\text{SPD}(u, v) := d(u, v) = \min_{\pi : u \rightsquigarrow v} |\pi|, \tag{23}$$

where $\pi$ ranges over all paths from $u$ to $v$ and $|\pi|$ denotes the number of edges along the path. This heuristic measures how many steps are required to connect $u$ and $v$ via the shortest route; if $u$ and $v$ are disconnected, then $d(u, v) = \infty$. Distances are normalized via the log transform $d \mapsto \log(1 + d)$. For disconnected pairs, a distinct penalty value $\log(1 + 1.5\, d_{\max})$ is assigned, where $d_{\max}$ is the maximum diameter observed among the connected components of the graph. No further percentile clipping is applied to SPD.

**PageRank.** Let $\mathbf{p} \in \mathbb{R}^{|\mathcal{V}|}$, where $\mathcal{V}$ is the node set, denote the PageRank stationary distribution (Brin & Page, 1998), defined as the solution to

$$\mathbf{p} = \alpha \, \mathbf{P}^\top \mathbf{p} + (1 - \alpha)\, \mathbf{v}, \tag{24}$$

where $\mathbf{P}$ is a random-walk transition matrix (e.g., $\mathbf{P} = \mathbf{D}^{-1}\mathbf{A}$ with degree matrix $\mathbf{D}$), $\alpha \in (0, 1)$ is the damping factor, and $\mathbf{v}$ is the teleportation distribution. PageRank quantifies node importance under this random-surfer model, and $p_u$ denotes the stationary probability assigned to node $u$. In our experiments, we use a uniform teleportation distribution and a damping factor of $\alpha = 0.85$. A symmetric pairwise score for $(u, v)$ is then given by their product $\text{PR}(u, v) := p_u\, p_v$. These pairwise scores can also be heavy-tailed, so the normalization protocol is applied similar to that for the Katz index: log-transformation $\log(\text{PR}(u, v) + \epsilon)$ with $\epsilon = 10^{-20}$, followed by z-score standardization, with upper-tail clipping at the 99$^{\text{th}}$ percentile.

### C.2. Real-world Experiments

We provide additional training details for the results presented in Table 1 and Table 2. For these experiments, we employed a minimal hyperparameter tuning strategy, focusing primarily on model depth and sampling configurations to ensure adequate capture of local neighborhood information. Additionally, we freeze the input projection matrix $\mathbf{W}_0$ across all

*Table 6.* Hyperparameter configurations of PENCIL for the HeaRT benchmark settings.

| Hyperparameter | cora | citeseer | pubmed | ogbl-collab | ogbl-ppa | ogbl-citation2 | ogbl-ddi |
|---|---|---|---|---|---|---|---|
| Sampling Configuration | (2, 20) | (2, 20) | (2, 20) | (1, 75) | (1, 150) | (1, 40) | (1, 350) |
| Hidden Size | 512 | 1024 | 512 | 512 | 512 | 512 | 512 |
| Intermediate Size | 2048 | 2048 | 2048 | 2048 | 2048 | 2048 | 2048 |
| # Layers | 4 | 2 | 4 | 8 | 4 | 3 | 8 |
| # Attention Heads | 8 | 8 | 8 | 8 | 8 | 8 | 8 |
| Effective Batch Size | 2048 | 2048 | 2048 | 8192 | 16384 | 65536 | 4096 |
| Learning Rate | 1.0E-04 | 1.0E-04 | 1.0E-04 | 1.0E-04 | 1.0E-04 | 1.0E-04 | 1.00E-04 |
| Weight Decay | 0.01 | 0.01 | 0.01 | 0.01 | 0.01 | 0.01 | 0.01 |
| # Epochs | 150 | 300 | 250 | 10 | 15 | 0.5 | 8 |

*Table 7.* Results on the original benchmark datasets. N/A means results are not available. Colored results indicate the 1st, 2nd, and 3rd-best performance in the corresponding metric. Category markers: ● Pure Heuristic, ● Heuristic-informed, ● ID-based, and ● Heuristic-agnostic + ID-free.

| Type | Model | cora | citeseer | pubmed | ogbl-collab | ogbl-ppa | ogbl-citation2 |
|---|---|---|---|---|---|---|---|
| | | MRR | MRR | MRR | H@50 | H@100 | MRR |
| ● | CN | 20.99 ± 0.00 | 28.34 ± 0.00 | 14.02 ± 0.00 | 61.37 ± 0.00 | 27.65 ± 0.00 | 74.30 ± 0.00 |
| | AA | 31.87 ± 0.00 | 29.37 ± 0.00 | 16.66 ± 0.00 | 64.17 ± 0.00 | 32.45 ± 0.00 | 75.96 ± 0.00 |
| | RA | 30.79 ± 0.00 | 27.61 ± 0.00 | 15.63 ± 0.00 | 63.81 ± 0.00 | 49.33 ± 0.00 | 76.04 ± 0.00 |
| ● | SEAL | 26.69 ± 5.89 | 39.36 ± 4.99 | 38.06 ± 5.18 | 63.37 ± 0.69 | 48.80 ± 5.61 | 86.93 ± 0.43 |
| | Neo-GNN | 22.65 ± 2.60 | 53.97 ± 5.88 | 31.45 ± 3.17 | 66.13 ± 0.61 | 48.45 ± 1.01 | 83.54 ± 0.32 |
| | BUDDY | 26.40 ± 4.40 | 59.48 ± 8.96 | 23.98 ± 5.11 | 64.59 ± 0.46 | 47.33 ± 1.96 | 87.86 ± 0.18 |
| | NCN | 32.93 ± 3.80 | 54.97 ± 6.03 | 35.65 ± 4.60 | 63.86 ± 0.51 | 62.63 ± 1.15 | 89.27 ± 0.05 |
| | NCNC | 29.01 ± 3.83 | 64.03 ± 3.67 | 25.70 ± 4.48 | 65.97 ± 1.03 | 62.61 ± 0.76 | 89.82 ± 0.43 |
| | LPFormer | 39.42 ± 5.78 | 65.42 ± 4.65 | 40.17 ± 1.92 | 68.14 ± 0.51 | 63.32 ± 0.63 | 89.81 ± 0.13 |
| ● | MPLP+ | N/A | N/A | N/A | 66.99 ± 0.40 | 65.24 ± 1.50 | 90.72 ± 0.12 |
| | Refined-GAE | N/A | N/A | N/A | 66.11 ± 0.35 | 78.41 ± 0.83 | 88.74 ± 0.06 |
| ● | GCN | 32.50 ± 6.87 | 50.01 ± 6.04 | 19.94 ± 4.24 | 54.96 ± 3.18 | 29.57 ± 2.90 | 84.85 ± 0.07 |
| | SAGE | 37.83 ± 7.75 | 47.84 ± 6.39 | 22.74 ± 5.47 | 59.44 ± 1.37 | 41.02 ± 1.94 | 83.06 ± 0.09 |
| | NBFNet | 37.69 ± 3.97 | 38.17 ± 3.06 | 44.73 ± 2.12 | N/A | N/A | N/A |
| | PENCIL w/o Features (Ours) | 42.23 ± 1.98 | 47.51 ± 3.09 | 38.28 ± 2.59 | 66.88 ± 0.34 | 73.85 ± 0.40 | 86.74 ± 0.26 |
| | PENCIL (Ours) | 32.12 ± 3.04 | 43.74 ± 5.47 | 38.34 ± 5.14 | 66.56 ± 0.19 | 79.54 ± 0.07 | 86.86 ± 0.20 |

runs, as we observed no significant impact on performance when it was learnable. Batch sizes were scaled according to hardware memory constraints. As detailed in Table 5 and Table 6, configurations are largely consistent across datasets; the notable exception is `citeseer`, which utilizes a larger hidden dimension and a reduced depth of two layers. For a fair comparison with the original study (Li et al., 2023), we use a single negative link for each positive link for all datasets. For `ogbl-ppa`, the HeaRT evaluation protocol (Li et al., 2023) is computationally prohibitive due to the customized negative links per positive edge, as discussed in Section 5. Consequently, only for this dataset, we utilized the optimal checkpoint identified in the original benchmark setting for the final HeaRT evaluation, leveraging the fact that training splits are identical across settings. All experiments were parallelized across 4/8 GPUs, utilizing NVIDIA A5000/A6000 cards for Planetoid benchmarks (Yang et al., 2016) and A100 cards for OGB benchmarks (Hu et al., 2020). Table 7 and Table 8 supplement the experimental results provided in the main text, where CN, AA, and RA are added for completeness.

## C.3. Ablation Study on Multiplicative Residual Connection

We perform an ablation study to isolate the contribution of the **multiplicative residual** defined in Section 3. Contemporary approaches often attempt to inject graph inductive biases into plain Transformers solely through PEs/SEs (Dwivedi & Bresson, 2021; Sanford et al., 2024; Yehudai et al., 2026; Ma et al., 2025a). However, Table 9 reveals that relying on input encodings alone is sub-optimal; the inclusion of the multiplicative residual provides substantial performance improvements across all datasets, indicating that an explicit structural prior is necessary for every layer.

*Table 8.* Results under HeaRT. N/A means results are not available. Colored results indicate the 1st, 2nd, and 3rd-best performance in MRR. Category markers: ● Pure Heuristic, ● Heuristic-informed, ● ID-based, and ● Heuristic-agnostic + ID-free.

| Type | Model | cora | citeseer | pubmed | ogbl-collab | ogbl-ppa | ogbl-ddi | ogbl-citation2 |
|------|-------|------|----------|--------|-------------|----------|----------|----------------|
| ● | CN | 9.78 ± 0.00 | 8.42 ± 0.00 | 2.28 ± 0.00 | 4.20 ± 0.00 | 25.70 ± 0.00 | 6.71 ± 0.00 | 17.11 ± 0.00 |
| | AA | 11.91 ± 0.00 | 10.82 ± 0.00 | 2.63 ± 0.00 | 5.07 ± 0.00 | 26.85 ± 0.00 | 6.97 ± 0.00 | 17.83 ± 0.00 |
| | RA | 11.81 ± 0.00 | 10.84 ± 0.00 | 2.47 ± 0.00 | 6.29 ± 0.00 | 28.34 ± 0.00 | 8.70 ± 0.00 | 17.79 ± 0.00 |
| ● | SEAL | 10.67 ± 3.46 | 13.16 ± 1.66 | 5.88 ± 0.53 | 6.43 ± 0.32 | 29.71 ± 0.71 | 9.99 ± 0.90 | 20.60 ± 1.28 |
| | Neo-GNN | 13.95 ± 0.39 | 17.34 ± 0.84 | 7.74 ± 0.30 | 5.23 ± 0.90 | 21.68 ± 1.14 | 10.86 ± 2.16 | 16.12 ± 0.25 |
| | BUDDY | 13.71 ± 0.59 | 22.84 ± 0.36 | 7.56 ± 0.18 | 5.67 ± 0.36 | 27.70 ± 0.33 | 12.43 ± 0.50 | 19.17 ± 0.20 |
| | NCN | 14.66 ± 0.95 | 28.65 ± 1.21 | 5.84 ± 0.22 | 5.09 ± 0.38 | 35.06 ± 0.26 | 12.86 ± 0.78 | 23.35 ± 0.28 |
| | NCNC | 14.98 ± 1.00 | 24.10 ± 0.65 | 8.58 ± 0.59 | 4.73 ± 0.86 | 33.52 ± 0.26 | N/A | 19.61 ± 0.54 |
| | LPFormer | 16.80 ± 0.52 | 26.34 ± 0.67 | 9.99 ± 0.52 | 7.62 ± 0.26 | 40.25 ± 0.24 | 13.20 ± 0.54 | 24.70 ± 0.55 |
| ● | MPLP+ | N/A | N/A | N/A | 6.79 | 41.40 | N/A | 23.11 |
| ● | GCN | 16.61 ± 0.30 | 21.09 ± 0.88 | 7.13 ± 0.27 | 6.09 ± 0.38 | 26.94 ± 0.48 | 13.46 ± 0.34 | 19.98 ± 0.35 |
| | SAGE | 14.74 ± 0.69 | 21.09 ± 1.15 | 9.40 ± 0.70 | 5.53 ± 0.50 | 27.27 ± 0.30 | 12.60 ± 0.72 | 22.05 ± 0.12 |
| | NBFNet | 13.56 ± 0.58 | 14.29 ± 0.80 | N/A | N/A | N/A | N/A | N/A |
| | PENCIL w/o Features (Ours) | 14.63 ± 0.52 | 16.50 ± 0.31 | 7.05 ± 0.17 | 5.25 ± 0.01 | 44.57 ± 0.15 | 14.07 ± 0.24 | 23.36 ± 0.07 |
| | PENCIL (Ours) | 13.13 ± 0.53 | 16.80 ± 1.43 | 8.88 ± 0.49 | 5.40 ± 0.05 | 45.43 ± 0.31 | N/A | 23.43 ± 0.14 |

*Table 9.* Ablation study on Multiplicative Residual (MR).

| Model | cora (2, 20) | | | pubmed (2, 16) | | | ogbl-collab (1, 75) | | |
|-------|------|------|------|------|------|------|------|------|------|
| | MRR | H@3 | H@20 | MRR | H@3 | H@20 | H@20 | H@50 | H@100 |
| PENCIL \MR | 34.64 ± 1.99 | 35.98 ± 3.97 | 58.67 ± 2.57 | 21.49 ± 4.30 | 21.81 ± 2.81 | 31.81 ± 2.32 | 49.52 ± 3.01 | 53.43 ± 1.92 | 56.04 ± 1.24 |
| PENCIL | 42.23 ± 1.98 | 43.34 ± 1.03 | 67.32 ± 3.56 | 38.28 ± 2.59 | 43.61 ± 1.48 | 57.49 ± 0.94 | 53.47 ± 0.85 | 66.88 ± 0.34 | 69.75 ± 0.45 |
| Performance Gain | +7.59 | +7.36 | +8.65 | +16.79 | +21.80 | +25.68 | +3.95 | +13.45 | +13.71 |

## C.4. Effects of Input Projection Matrix Initialization

In this section, we study how the initialization of the input projection matrix $\mathbf{W}_0$ influences optimization and performance. We compare (i) **orthogonal** initialization (our default), (ii) a **mean-shifted Gaussian** initialization with mean 0.1 to probe sensitivity to non-zero-mean embeddings, and (iii) a **low-rank** initialization with rank 5, which constrains the embedding space and limits the number of mutually orthogonal directions. Following the same implementation as discussed earlier, we freeze $\mathbf{W}_0$. This also isolates the initialization scheme as the sole variable affecting optimization. All experiments use the same hyperparameters as Table 5. Figure 7 reports test MRR averaged over 5 runs on citeseer, cora, and pubmed. Orthogonal initialization yields the fastest and most stable convergence and achieves the strongest final MRR across datasets, whereas low-rank initialization substantially degrades performance and converges to markedly lower MRR. The mean-shifted Gaussian initialization exhibits noticeably higher variance and slower, less stable training. We also evaluate adding learnable word position embeddings (WPE). Across all settings, WPE induces at most minor and dataset-dependent differences and does not consistently improve either convergence or final MRR, suggesting that—under this graph input encoding—sequential positional inductive biases are irrelevant for performance.

## C.5. Effect of Multiple Labeled Subgraphs

We further examine whether using multiple labeled subgraphs per candidate link improves performance. Increasing the number of such subgraphs can substantially increase memory cost, since each additional subgraph requires an additional forward pass. In Table 10, we compare PENCIL with a variant that applies DeepSets-style pooling (Zaheer et al., 2017) over three labeled subgraphs per sample following the graph encoding scheme in Section 3.1. We do not observe clear performance gains from using multiple labeled subgraphs. Moreover, as noted by Zhou et al. (Zhou et al., 2023), selecting multiple labeled subgraphs often requires additional node-selection policies, introducing further computational overhead.

## C.6. Link Prediction on Latent Space Graphs

In this section, we further evaluate PENCIL and several simple baselines on synthetic latent-space graphs (Hoff et al., 2002). This controlled setting allows us to separately vary the strength of structural signal and node-feature signal for link prediction.

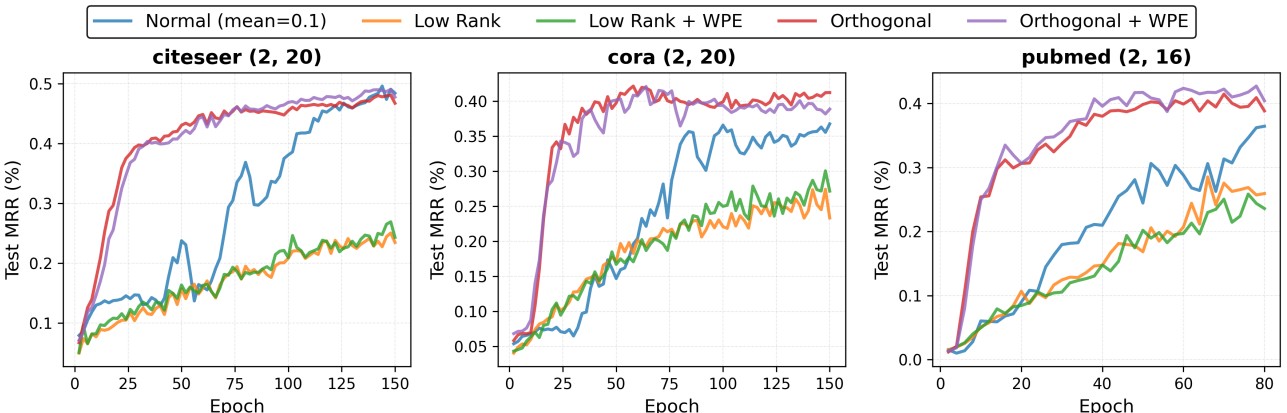

*Figure 7.* MRR averaged over 5 runs on the `citeseer`, `cora`, and `pubmed` datasets using different input embeddings' initializations.

*Table 10.* Effect of using multiple labeled subgraphs. We compare PENCIL with a variant that applies DeepSets-style pooling over three labeled subgraphs per sample. Higher values are better.

| Model | cora (2, 20) | citeseer (2, 20) | ogbl-collab (1, 75) |
|---|---|---|---|
| | MRR | MRR | H@50 |
| PENCIL | $42.23 \pm 1.98$ | $47.51 \pm 3.09$ | $66.88 \pm 0.34$ |
| PENCIL + LRP-3 | 39.18 | 42.17 | 67.05 |

We generate a latent-space graph with $N$ nodes and $K$ clusters as follows. Each node $i$ is first assigned a cluster label $y_i$. Conditional on this label, its latent position is sampled from a Gaussian distribution:

$$\mathbf{z}_i \mid y_i \sim \mathcal{N}(\mathbf{c}_{y_i}, \sigma_{\text{noise}}^2 I),$$

where the cluster centers $\mathbf{c}_k \in \mathbb{R}^{d_{\text{latent}}}$ are placed along coordinate axes with magnitude $s$. Given the latent positions, each undirected edge is independently sampled according to

$$A_{ij} \sim \text{Bernoulli}\left(\sigma\left(\alpha - \beta \|\mathbf{z}_i - \mathbf{z}_j\|_2^2\right)\right),$$

where $\alpha$ controls the overall graph density and $\beta$ controls the sensitivity of edge formation to latent distance.

Node features are generated as noisy random projections of the latent positions. Let $\mathbf{W} \in \mathbb{R}^{d_{\text{latent}} \times p}$ be a random projection matrix with entries $W_{ab} \sim \mathcal{N}(0, 1/p)$. The feature vector of node $i$ is then given by

$$\mathbf{x}_i = \sqrt{\mu}\,\mathbf{z}_i \mathbf{W} + \boldsymbol{\eta}_i, \qquad \boldsymbol{\eta}_i \sim \mathcal{N}(\mathbf{0}, I_p/p).$$

Here, $\mu$ controls the signal-to-noise ratio of the node features: larger values of $\mu$ make the observed features more informative about the latent position.

Under this generative process, the cluster label $y_i$ determines the distribution of the latent position $\mathbf{z}_i$, and $\mathbf{z}_i$ subsequently determines both the edge probabilities and the node features. Thus, structural information and node-feature information are both induced by the same latent geometry. The parameter $s$ controls the separation among cluster centers, thereby modulating the strength of latent structural homophily, while $\mu$ controls how strongly the latent positions are reflected in the observed node features.

In our experiments, we fix $N = 5000$, $K = d_{\text{latent}} = 3$, $p = 32$, $\beta = 2$, and $\sigma_{\text{noise}} = 0.5$, and vary only $s$ and $\mu$. For each configuration, we dynamically adjust $\alpha$ so that the generated graph has average degree approximately 15. We use 90% of observed positive edges for training and hold out the remaining 10% for testing. For evaluation, we construct a balanced link-prediction dataset by sampling an equal number of positive and negative node pairs. Negative pairs are selected from non-edges whose endpoints are far apart in latent space, yielding a controlled setting in which positive and negative examples are well separated by latent distance. Figures 8 and 9 visualize the latent geometry and distance distributions under our experimental settings.

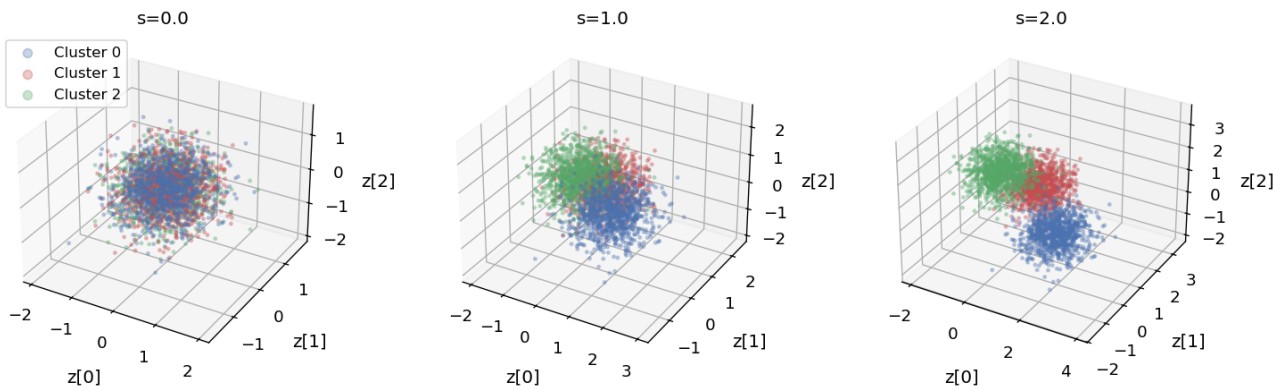

*Figure 8.* Latent geometry under varying cluster separation $s$.

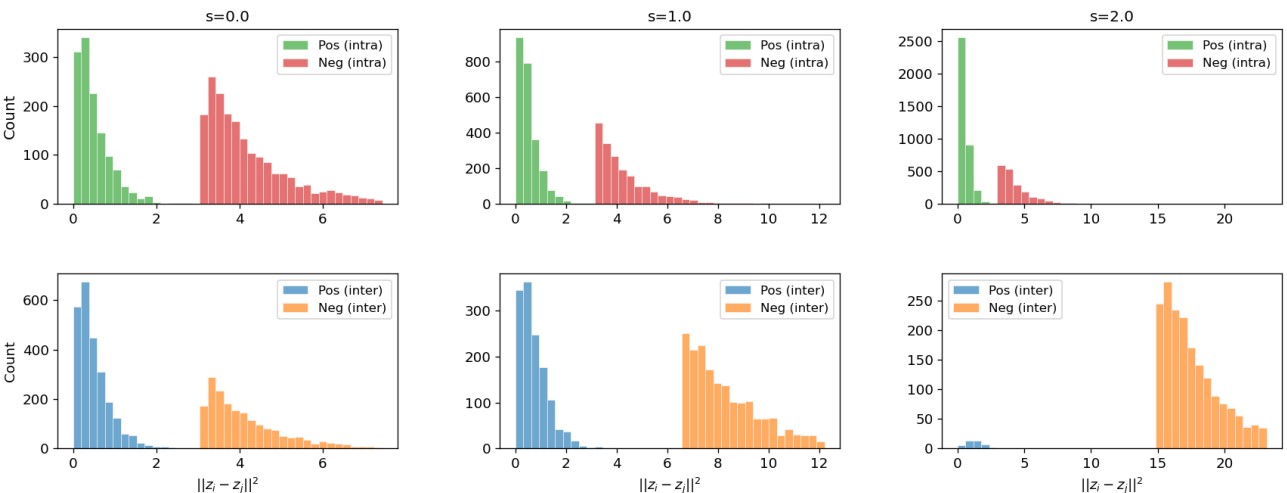

*Figure 9.* Squared latent-distance distributions for positive and negative intra-/inter-cluster pairs under varying $s$.

We evaluate two experimental axes: varying the structural signal $s$ while fixing $\mu = 1$, and varying the node-feature signal $\mu$ while fixing $s = 1$. We additionally consider two edge cases, $(s = 2, \mu = 0)$ and $(s = 0, \mu = 8)$, corresponding to strong structural signal without informative node features and strong node-feature signal without cluster separation, respectively. Larger values of $s$ and $\mu$ indicate stronger structural and node-feature signals. All neural models use two layers with hidden dimension fixed to 256. The experiments are reported in Table 11 and 12.

We make several observations. First, pairwise structural heuristics such as CN and AA perform poorly across all settings, indicating that local overlap-based scores alone are insufficient to recover the latent-distance signal. Second, neural models benefit substantially from informative node features, with performance generally improving as $\mu$ increases. Third, although GAT and PENCIL are both attention-based models, their attention mechanisms use fundamentally different inputs. GAT computes attention weights for feature aggregation over local neighborhoods, so its attention scores are directly tied to node-feature quality; consequently, its performance drops sharply when the features are uninformative ($\mu = 0$). In contrast, PENCIL applies self-attention over a tokenized link-centric subgraph whose tokens explicitly encode the graph structure. This allows PENCIL to attend over pairwise relationships among nodes in the sampled subgraph, rather than relying only on feature-based neighbor weighting. Finally, PENCIL achieves the best performance in most settings, suggesting that it can effectively leverage node-feature cues when they are informative while retaining access to structural cues through its subgraph tokenization.

*Table 11.* Link-prediction performance on latent-space graphs when varying the structural signal $s$ while fixing $\mu = 1$, with the additional edge case $(s, \mu) = (2, 0)$. Reported metric is MRR.

| Model | $(s = 0, \mu = 1)$ | $(s = 1, \mu = 1)$ | $(s = 2, \mu = 1)$ | $(s = 2, \mu = 0)$ |
|---|---|---|---|---|
| CN | 1.58 | 4.78 | 4.48 | 4.48 |
| AA | 2.85 | 5.43 | 7.37 | 7.37 |
| GCN | 75.48 | 70.08 | 83.68 | **40.67** |
| GAT | 71.44 | **81.29** | 82.11 | 6.27 |
| PENCIL | **77.63** | 74.37 | **93.59** | 36.01 |

*Table 12.* Link-prediction performance on latent-space graphs when varying the node-feature signal $\mu$ while fixing $s = 1$, with the additional edge case $(s, \mu) = (0, 8)$. Reported metric is MRR.

| Model | $(s = 1, \mu = 0)$ | $(s = 1, \mu = 2)$ | $(s = 1, \mu = 8)$ | $(s = 0, \mu = 8)$ |
|---|---|---|---|---|
| CN | 4.78 | 4.78 | 4.78 | 1.58 |
| AA | 5.43 | 5.43 | 5.43 | 2.85 |
| GCN | 10.09 | 71.04 | 73.85 | 77.05 |
| GAT | 5.36 | 83.57 | 83.04 | 76.01 |
| PENCIL | **17.21** | **84.90** | **92.05** | **95.34** |

### C.7. Computational Complexity

In this section, we analyze the computational complexity of PENCIL and GAT under three perspectives: batching, forward pass, and node labeling overhead. Our goal is to show that PENCIL remains computationally efficient across these dimensions, owing to its NLP-style batching scheme, efficient attention kernels, and labeling-free architecture.

#### C.7.1. BATCHING OVERHEAD

We benchmark the overhead of constructing mini-batches of *isolated* sampled subgraphs and the associated peak RAM usage, comparing a padding-and-stacking implementation based on native tensor operations against PyG collation routines (Fey et al., 2025; Fey & Lenssen, 2019). This setting differs from PyG's neighbor-sampling loaders (e.g., `NeighborLoader` and `LinkNeighborLoader`), which typically construct a single computation graph for a batch of seed nodes/edges and thus *merge* overlapping neighborhoods into one (potentially larger) subgraph. In contrast, PENCIL operates on a fixed-budget collection of independent subgraph instances; even if two samples share underlying nodes in the original graph, they are treated as separate instances after sampling. Since this isolated subgraph list representation is not the typical output of PyG's neighbor-sampling loaders, we use PyG's `from_data_list` as a GNN-style batching baseline to collate a list of variable-sized `Data` objects extracted by the ShaDowKHop sampler (Zeng et al., 2021), and compare it against a Transformer-style batching procedure that pads each subgraph to a fixed budget and stacks the resulting tensors. Figure 10 reports batching time and peak RAM on `ogbl-collab` as a function of batch size. The padding-and-stacking approach yields substantially lower batching overhead and consistently low peak RAM, whereas `from_data_list` incurs significantly higher batching time and rapidly increasing memory consumption as batch size grows. We attribute this gap primarily to implementation-level batching overhead: padding-and-stacking produces contiguous dense tensors with a fixed shape, while `from_data_list` must concatenate many variable-sized graph objects (including index offsetting and allocation of batched sparse structures), leading to higher collation cost and memory footprints that scale with the total number of nodes/edges in the batch.

#### C.7.2. TRAINING/INFERENCE TIME

We benchmark per-mini-batch training and inference time for PENCIL and GAT on `pubmed` and `ogbl-collab` under identical sampling and model configurations. Using the same sampling configuration isolates how runtime depends on graph sparsity: `ogbl-collab` exhibits higher connectivity than `pubmed` (e.g., higher average degree; see Table 7 in (Li et al., 2023)). Table 13 reports wall-clock time (seconds; mean $\pm$ std) when varying depth. On `pubmed`, GAT is consistently faster than PENCIL for both training and inference. On `ogbl-collab`, however, the two models have comparable per-batch costs, with the gap substantially reduced. This trend is consistent with the underlying computation: GAT computes

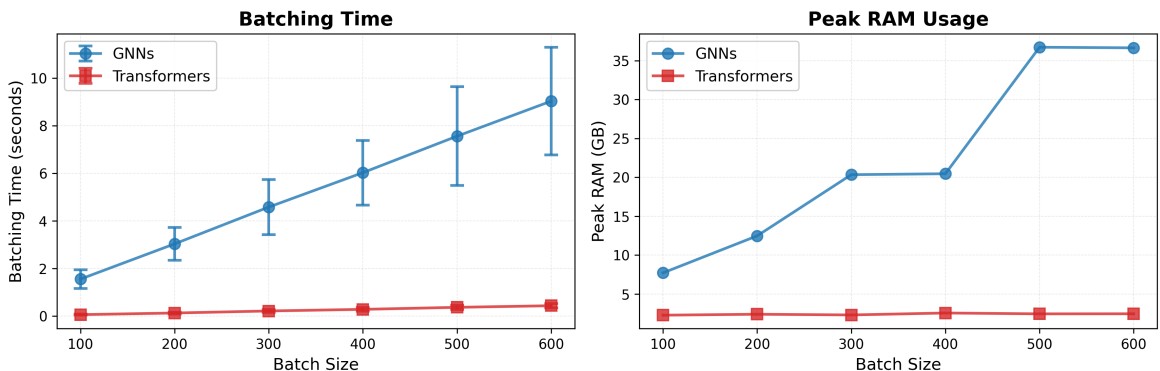

*Figure 10.* Batching time and memory comparison of Transformers and GNNs on the `ogbl-collab` dataset.

*Table 13.* Number of parameters and wall-clock time per mini-batch (seconds; mean $\pm$ std) for training and inference of PENCIL and GAT with 3 and 8 layers on `ogbl-collab` and `pubmed` under the $(1, 75)$ sampling configuration. All measurements are collected on the same hardware configuration, using batch size 100, and averaged over 50 batches.

| Model | # Params | ogbl-collab (1, 75) | | pubmed (1, 75) | |
|---|---|---|---|---|---|
| | | Training Time Per Batch | Inference Time Per Batch | Training Time Per Batch | Inference Time Per Batch |
| PENCIL-3L | 10.2M | 0.019 ± 0.045 | 0.012 ± 0.039 | 0.016 ± 0.036 | 0.010 ± 0.029 |
| PENCIL-8L | 27.3M | 0.035 ± 0.049 | 0.020 ± 0.043 | 0.029 ± 0.038 | 0.016 ± 0.031 |
| GAT-3L | 1.32M | 0.021 ± 0.078 | 0.015 ± 0.070 | 0.011 ± 0.033 | 0.007 ± 0.026 |
| GAT-8L | 3.95M | 0.032 ± 0.079 | 0.017 ± 0.047 | 0.020 ± 0.034 | 0.011 ± 0.027 |

attention only on sampled edges, so its cost grows with the number of edges present in sampled subgraphs, whereas PENCIL computes dense attention across all tokens with each sampled subgraph instance. Notably, despite having roughly 7–8× more parameters than GAT in these settings and using dense self-attention, PENCIL remains competitive, benefiting from highly optimized Transformer implementations (Wolf et al., 2020; Dao et al., 2022). Finally, linear-attention variants could further reduce the compute and memory footprint of self-attention during both training and inference for large token budgets (Choromanski et al., 2021); we leave this extension to future work.

### C.7.3. NODE LABELING OVERHEAD

We further measure the computational overhead of subgraph extraction under commonly used structural and positional encodings. Specifically, we evaluate extraction time and memory usage on a synthetic graph with 10,000 nodes and average degree 15, comparing unlabeled subgraphs with DRNL, Random Walk Positional Encoding (RWPE) of length 8, and Laplacian Positional Encoding (LapPE) with $k = 4$, which are commonly used in SEAL-style and graph Transformer methods. As shown in Table 14, PEs introduce substantial overhead, and even DRNL requires approximately $1.6\times$ to $4\times$ more extraction time than the unlabeled subgraphs used by PENCIL.

*Table 14.* Subgraph extraction overhead on a 10,000-node graph with average degree 15. Time is reported in milliseconds and memory is reported in megabytes.

| Sampling config | Maximum # nodes | No labeling | | DRNL | | RWPE | | LapPE | |
|---|---|---|---|---|---|---|---|---|---|
| | | Time | Mem. | Time | Mem. | Time | Mem. | Time | Mem. |
| $(1, 20)$ | 42 | 0.109 | 0.07 | 0.430 | 0.19 | 0.161 | 0.05 | 0.270 | 0.07 |
| $(1, 30)$ | 62 | 0.132 | 0.05 | 0.454 | 0.05 | 0.184 | 0.05 | 0.387 | 0.11 |
| $(2, 20)$ | 842 | 1.392 | 0.07 | 2.225 | 0.35 | 6.980 | 0.07 | 6.889 | 0.24 |
| $(2, 30)$ | 1862 | 4.155 | 0.17 | 5.676 | 1.10 | 60.371 | 0.17 | 20.846 | 0.50 |

