# OpenReview forum: "Plain Transformers are Surprisingly Powerful Link Predictors"
_ICML.cc/2026/Conference — ICML 2026 regular_

### Official Review · Reviewer_k3aV · 2026-02-27

**Soundness:** 4
**Presentation:** 2
**Significance:** 4
**Originality:** 4
**Overall Recommendation:** 4
**Confidence:** 5

**Summary:**

This paper proposes PENCIL, a link prediction model based on plain Transformers. For each candidate edge, a local subgraph is sampled with a fixed budget and modeled using a standard BERT-style encoder. The paper theoretically proves the expressiveness of PENCIL, and the idea of pairwise heuristic estimation is very good.

**Compliance With Llm Reviewing Policy:**

Affirmed.

**Final Justification:**

The paper proposes a simple and effective Transformer-based approach for link prediction, with a clear design and solid empirical results. The idea of modeling link-centric subgraphs with a plain Transformer is intuitive, and the heuristic estimation experiments are helpful.

There are still some limitations, such as the gap between theoretical results and experiments (e.g., LRP not evaluated), and the need to better clarify the role of heuristic estimation.

Overall, I find the paper technically sound and a meaningful contribution, and I support acceptance.

**Key Questions For Authors:**

1. Although the authors provide sufficient proof regarding expressive power, do some experimental results indicate that the current model's generalization ability is insufficient?

2. Does the sampling budget Nmax determine the upper limit of expressive power?

**Limitations:**

yes

**Strengths And Weaknesses:**

### Strengths

1. The design approach is clear and novel. The paper directly challenges current mainstream practices by proposing plain Transformer + subgraph tokenization, which is conceptually simple with well-motivated problem formulation.

2. The theoretical analysis is structurally complete, covering distributional permutation invariance under random canonicalization, reduction to path algorithms like Bellman-Ford/NBFNet, and many other aspects.

3. The pairwise heuristic estimation experiments are novel and valuable, explaining the capabilities of graph representation learning models from a structural understanding perspective.

### Weaknesses

1. Most theoretical conclusions are lower bounds, with many theoretical results taking the form that there exist parameter settings such that PENCIL can reduce to some model or algorithm. While I appreciate the design of the pairwise heuristic estimation task, this does not imply a direct connection to link prediction. The authors should more clearly explain their motivation for doing this.

2. In Theorem 4.9, the authors indicate that the PENCIL + LRP setting would be more expressive, but this setting does not seem to appear in the experiments. Why?

3. There are too many dashes in the paper.

---

> ### Author Rebuttal · Authors · 2026-03-31
>
> Thank you very much for your thoughtful review. We’d like to address the concerns.
>
> **[W1] Implications of pairwise heuristic estimation**
>
> We appreciate this constructive feedback and will clarify the connection between pairwise heuristics and link prediction more explicitly. First, prior work has established theoretical links between classical pairwise heuristics and link prediction, which explains why such simple heuristics can already be strong baselines [1]. Second, many state-of-the-art methods incorporate these heuristics as explicit input signals; in contrast, our result shows that a Transformer can estimate them directly from the input graph, without hard-coding them as features. This means the model can recover and exploit such structural cues natively during learning. For example, when a graph is biased toward common-neighbor patterns, PENCIL can degenerate to a variant that estimates common neighbors, as shown in the main text. Third, heuristic estimation serves as a controlled probe of structural understanding: unlike link prediction, which depends on many factors beyond topology, pairwise heuristics are determined by graph structure alone. They therefore provide a clean benchmark for assessing how well Transformers can recover fundamental structural signals, and where their limitations may lie.
>
> **[W2] PENCIL + LRP**
>
> As noted in the main text, we find that in practice PENCIL already performs strongly with just one labeled subgraph per sample. Increasing the number of labeled subgraphs substantially raises memory cost because it requires additional forward passes. For example, in preliminary result below, we apply set pooling over three labeled subgraphs per sample, yet do not observe clear evidence of performance gains.
>
> | Model | cora(2, 20) (MRR) | citeseer(2, 20) (MRR) | collab(1, 75) (H@50) |
> | --- | --- | --- | --- |
> | PENCIL | 42.23 ± 1.98 | 47.51 ± 3.09 | 66.88 ± 0.34 |
> | PENCIL + LRP-3 | 39.18 | 42.17 | 67.05 |
>
> **[W3] There are too many dashes.**
>
> We will replace the dashes in the final version appropriately.
>
> **[Q1] Although the authors provide sufficient proof regarding expressive power, do some experimental results indicate that the current model's generalization ability is insufficient?**
>
> We interpret the reviewer’s question about generalization as referring to the empirical performance reported in the paper. We would like to emphasize that greater theoretical expressiveness does not necessarily translate into stronger link prediction performance. (Lachi et al., 2025) has shown that more expressive models do not consistently dominate in practice, and their advantages often appear only in particular graph regimes, such as highly symmetric graphs. Moreover, expressiveness theory typically assumes access to the full graph, while practical link prediction operates under stricter constraints: GNNs are kept shallow, and in our setting the model is further restricted to fixed-budget sampled subgraphs. Finally, many strong baselines exploit global information that is deliberately unavailable to PENCIL, and therefore benefit from signals outside the local subgraph. Taken together, these factors make it unsurprising that strong theoretical guarantees do not always coincide with state-of-the-art empirical performance.
>
> **[Q2] Does the sampling budget Nmax determine the upper limit of expressive power?**
>
> Increasing the sampling budget $N_{\text{max}}$ can strengthen PENCIL’s expressive power, as proven by (Zhou et al., 2023), which shows that labeling more nodes in the subgraph yields strictly stronger expressiveness. Thus, sampling budget serves as a natural knob controlling the expressive power accessible in practice. More broadly, this suggests that advances in hardware and systems for processing larger sampled subgraphs can directly expand the practical capabilities of models like PENCIL.
>
> We hope our responses sufficiently address the reviewer’s concerns, and we kindly ask the reviewer to consider revising their evaluation if they find these clarifications satisfactory.
>
> [1] Theoretical Justification of Popular Link Prediction Heuristics.

---

> > ### Author Rebuttal · Reviewer_k3aV · 2026-04-01
> >
> > Thank you for your responses. I appreciate the clarifications.
> >
> > I encourage the authors to include LRP-related experiments in a future revision to better align the theoretical claims with empirical results, and to further refine the presentation (especially the connection between heuristic estimation and link prediction).
> >
> > I will maintain my current score.

---

> > > ### Author Response · Authors · 2026-04-06
> > >
> > > We appreciate the reviewer finding our paper and rebuttal compelling. We will ensure that all promised revisions are reflected in the final manuscript. Please let us know if you have any remaining questions regarding our work.

---

### Official Review · Reviewer_4Nws · 2026-03-10

**Soundness:** 3
**Presentation:** 3
**Significance:** 2
**Originality:** 1
**Overall Recommendation:** 3
**Confidence:** 5

**Summary:**

This paper discusses PENCIL, a link prediction approach based predominantly on the use of a plain transformer layer, with minimal graph-specific encodings. The approach extracts subgraphs around a source and target node, and uses a simple node encoding for both nodes and their surrounding context (ID + adjacency + role), effectively turning nodes into the transformer's tokens. The paper then discusses how this approach can actually recover many structural heuristics and metrics, and in doing so emulate (and even surpass) the performance of dedicated MPNNs. On the empirical front, the paper conducts an experimental study, showcasing that this minimalistic approach obtains strong performance, albeit with some important nuances around dataset size and training efficiency.

**Compliance With Llm Reviewing Policy:**

Affirmed.

**Final Justification:**

UPDATE: I have raised my score to 3 in line with the authors' argumentation during the rebuttal period, but I overall continue to lean against this work in its current form.

**Key Questions For Authors:**

None, please address my points in the weaknesses section above.

**Limitations:**

The authors discuss limitations of their work quite honestly and openly in the conclusion section.

**Strengths And Weaknesses:**

Strengths:
- Very intuitive idea that is easy to understand and follow
- Empirical analysis, while smaller, showcases the points of the paper quite well
- Interesting experiments comparing PENCIL to a set of established baselines on common graph metrics.

Weaknesses:
- Unfortunately, I do not think PENCIL offers any novel insights or useful takeaways that would benefit the literature in this space. Arguably, observing that transformers with minimal graph inductive biases can perform competitively is a useful contribution, but I found myself struggling to see where to go next with this model after reading the paper. It is not particularly clear which scenarios warrant using a PENCIL-style model, what should be done to make them more viable, and what makes this approach stand out. As a result, I strongly recommend that the authors of this paper add further take-away messages (even if negative or instructional) that could benefit the work. For example, a clearer set of guidelines on the trade-offs of classical transformers (i.e., PENCIL) versus the other approaches is a good starting point, but ideally, I would suggest that the authors delve deeper into more foundational contributions explaining why the loss of structural features was not particularly detrimental (and even beneficial in some cases) to performance.

- The paper does not sufficiently explore the aspects that are actually intriguing about its results. In particular, the fact a transformer can emulate more specialized models' performance, in my view, is not the result, but rather the preamble to a deeper analysis. In and of itself, the former observation can be expected in some scenarios, given the success of graph transformers and similar setups in the literature. Yet, a deeper exploration into *why* such a result holds and what we can learn from it would be especially valuable. As such, I strongly encourage the authors to emphasize the technical design of PENCIL less, and instead focus more on experiments that highlight the key successes / failures of their approach: Feature ablations are a good step in this direction, as are explorations on synthetic datasets of varying underlying graph structures.

- As a more minor weakness, I don't think this particular approach would be very scalable for deployment given that it requires subgraph sampling and encoding for every single edge being predicted. The "deployability" aspect is mentioned at various points, most notably in the motivation section. PENCIL does address many of the points in that motivation, but also introduces a subgraph sampling step, which I believe should be acknowledged and tested (runtime, space requirements) more rigorously.

---

> ### Author Rebuttal · Authors · 2026-03-31
>
> We thank you for your constructive feedback. We hope our responses address your concerns and merit a higher score.
>
>
> **Significance and Originality**
>
> We appreciate this thoughtful concern. We respectfully clarify that the central contribution of PENCIL is **not simply that a Transformer can be competitive for link prediction**, but that we provide, the first systematic explanation of **what enables a vanilla bidirectional Transformer to do so without the strong positional encodings (PEs) typically required by graph transformers (GTs)**.
>
> Our primary contribution, along with the model design, is the paper’s foundational analysis on distributional permutation invariance, generalization of pairwise heuristics, and expressiveness in the link prediction setting. These results are non-trivial precisely because they hold **without the PE assumptions on which traditional GTs usually rely**. We also emphasize that these insights extend beyond PENCIL itself. They apply to all bidirectional-attention Transformers using the same input encoding scheme whenever graph structure can be materialized through attention.
>
> **Usefulness of structural/node features and ablation studies on synthetic data**
>
> We agree that this is an important question. The broader interplay among node features, graph structure, and downstream tasks is a longstanding challenge in graph ML, whereas our focus here is understanding Transformers from a model-centric perspective. Prior work has shown that the usefulness of node and structural features is highly task- and dataset-dependent [1], and has benchmarked these effects from both model-centric and model-agnostic viewpoints [2]. Our remark of node-feature usefulness is therefore model-centric, based on performance changes after ablating node attributes.
>
> The remark doesn’t mean PENCIL can’t leverage node features as suggested by drops in performance on some datasets. We ran a controlled synthetic experiment on latent-space graphs [3] to highlight that PENCIL can leverage node feature. Due to space constraints, we provide additional details in our response to Reviewer yRG3 (Q1), who raised a similar concern. In other words, we believe this apparent incompatibility is primarily a **data-regime issue rather than a model limitation of PENCIL**.
>
> **Scalability and deployability**
>
> Our deployment setting is realistic because many real-world link prediction systems explicitly include subgraph sampling [4]. PENCIL's main scalability challenge arises at online inference, where link representations must be computed in real time. When the candidate set is small, this cost can be mitigated through distributed inference, though we agree this is not a sustainable long-term solution. Developing a more scalable variant of PENCIL that avoids edge-centric subgraph sampling is therefore an important direction for future work.
>
> We also measure subgraph extraction time (ms) and memory (MBs) on a 10,000-node graph (average degree 15), with and without DRNL, RWPE (length 8), and LapPE (k=4), which are commonly used in SEAL and GTs. PEs add substantial overhead, and even DRNL costs roughly $1.6\times$  to $4\times$ more than PENCIL’s unlabeled small graphs.
>
> | **Sampling config** | **Maximum # nodes** | **No Labeling** | **DRNL** | **RWPE** | **LapPE** |
> | --- | --- | --- | --- | --- | --- |
> | (1, 20) | 42 | 0.109/0.07 | 0.43/0.19 | 0.161/0.05 | 0.27/0.07 |
> | (1, 30) | 62 | 0.132/0.05 | 0.454/0.05 | 0.184/0.05 | 0.387/0.11 |
> | (2, 20) | 842 | 1.392/0.07 | 2.225/0.35 | 6.98/0.07 | 6.889/0.24 |
> | (2, 30) | 1862 | 4.155/0.17 | 5.676/1.10 | 60.371/0.17 | 20.846/0.50 |
>
> **Useful takeaways**
>
> We believe PENCIL points to a new paradigm for Transformer-based randomized link predictors. We show that a vanilla bidirectional Transformer over sampled local subgraphs can internally estimate classical pairwise heuristics and retain key theoretical properties associated with GNN-based link predictors, even without dedicated PEs. In practice, PENCIL can be deployed similarly to inductive GNNs, without requiring global embeddings or handcrafted pairwise heuristics as other baselines; its main trade-off is edge-centric subgraph sampling at inference time. At the same time, we highlight the core bottleneck of this paradigm: its expressiveness and generalization depend strongly on the sampled subgraph size, which limits scalability. We therefore view the broader takeaway as follows: minimal graph-specific bias can already deliver strong large-scale link prediction, and the next frontier is to retain this simplicity while breaking the subgraph-scaling bottleneck.
>
> [1] Oversmoothing, "Oversquashing", Heterophily, Long-Range, and more: Demystifying Common Beliefs in Graph Machine Learning.
>
> [2] No Metric to Rule Them All: Toward Principled Evaluations of Graph-Learning Datasets.
>
> [3] Latent Space Approaches to Social Network Analysis.
>
> [4] GiGL: Large-Scale Graph Neural Networks at Snapchat

---

> > ### Author Rebuttal · Reviewer_4Nws · 2026-04-04
> >
> > I thank the authors for their clarifications. I recognize the difference in setting, and appreciate that there is a non-trivial effort to produce results in this work. I therefore will raise my score to a 3. However, I still believe that this work requires a deeper analysis per my second point, which would not be feasible within this rebuttal period.

---

> > > ### Author Response · Authors · 2026-04-06
> > >
> > > We sincerely thank the reviewer for engaging with our rebuttal, and we'd like to address the remaining concerns as directly as possible.
> > >
> > > **Novelty and significance.** Our paper studies a previously underexplored question: whether a vanilla Transformer, instantiated in our work as a BERT-based randomized predictor, is fundamentally capable of strong link prediction without relying on carefully designed positional encodings (PEs). In particular, we demonstrate that plain Transformers can estimate a broad class of classical pairwise heuristics and, in turn, serve as strong link predictors in practice. This is a non-trivial finding. Beyond empirical evidence, our theoretical contribution is comprehensive: we study three complementary properties central to graph learning models: (i) distributional permutation invariance, (ii) pairwise heuristic estimation, and (iii) expressiveness. These properties are often analyzed in isolation, whereas our work unifies them in a single framework for Transformer-based link prediction. In this sense, our contribution is not an incremental extension of existing Graph Transformer literature, but a principled step toward establishing **plain Transformers as a legitimate and theoretically grounded paradigm** for link prediction, with broader implications for graph machine learning. We further strengthen this perspective by connecting our model to foundational works such as NBFNet and MPLP, which helps situate PENCIL within link prediction methods.
> > >
> > >
> > > **Scope.** Our paper is intentionally **model-centric**. The main goal is to identify and analyze the three properties mentioned in **Novelty and significance** that make Transformers effective link predictors. The feature ablation in the main paper is included only as a preliminary study of the interaction among node features, structural signals, and the task itself on standard benchmarks. Its role is not to provide a full data-centric characterization of link prediction datasets, but to show that structural information alone is often already sufficient in many common settings. While we agree that a deeper investigation of the feature–structure–task interplay is valuable, it is out of the main focus of this work and we will explore it as a promising future research direction.
> > >
> > > **Feature ablation on synthetic data.** In direct response to the reviewer’s suggestion, we conducted a substantially more controlled analysis on synthetic graphs generated from a latent-space graph model (see the Q1 experiment for Reviewer yRG3). These experiments vary the informativeness of structural and feature signals from non-informative to fully descriptive, allowing us to examine model behavior across a broad spectrum of regimes. The results consistently show that PENCIL is able to exploit whatever predictive information is available and outperform both GNN baselines and classical pairwise heuristics across all settings we tested. We believe this directly addresses the request for deeper analysis on feature ablation. While we welcome more concrete suggestions and more specific phenomena for future investigation, we respectfully note that we have already carried out the additional experiments requested and that they strongly support our claims.
> > >
> > >
> > > **Trade-offs relative to GNNs.** We have clearly articulated, in both the main paper and the rebuttal, how PENCIL differs from existing GNNs in terms of theoretical properties (Section 4), performance (Section 5), and computational characteristics (Appendix C.5 and the rebuttal). The overall picture is consistent: PENCIL is not only competitive in accuracy, but also practically scalable relative to relevant subgraph-based/node labeling baselines, particularly when accounting for the costs of node labeling and subgraph processing. Importantly, because PENCIL’s capability improves with larger sampled subgraphs, advances in hardware can directly expand its practical potential.
> > >
> > > **Scalability.** We have been fully transparent that the principal limitation of PENCIL lies in edge-centric subgraph sampling, which is explicitly discussed in the main text, and we identify it as the primary direction for future work. At the same time, this limitation should be viewed in context. First, this limitation can be addressed effectively by parallelization. Second, our analysis already shows that PENCIL remains competitive with lightweight GNNs in batching, training, and inference time, while substantially outperforming alternative node-labeling strategies in computational overhead. Thus, the practical trade-off is clear: PENCIL introduces a sampling bottleneck, but in return offers strong predictive power, principled theoretical grounding, and a favorable scalability profile proven to scale industry.
> > >
> > > We hope these final clarifications help resolve the remaining concerns and better reflect the strengths of the work. We would be grateful if the reviewer could take these clarifications into account in their final assessment.

---

### Official Review · Reviewer_yRG3 · 2026-03-10

**Soundness:** 3
**Presentation:** 3
**Significance:** 3
**Originality:** 4
**Overall Recommendation:** 4
**Confidence:** 4

**Summary:**

This paper proposes PENCIL, an encoder-only plain Transformer for graph link prediction, addressing GNN’s structural expressivity limitations and node-centric aggregation flaws, as well as existing Graph Transformers’ heavy overhead from complex structural/positional encodings and violation of deployment constraints, along with ID-based methods’ poor scalability and high parameter redundancy. This paper designs a link-centric local subgraph encoding scheme to replace traditional structural/positional encodings, enabling direct adjacency matrix reconstruction from input encodings without offline computation or full-graph structure. This paper adds a multiplicative residual connection to the standard BERT-style encoder, fusing bidirectional self-attention outputs with adjacency matrix-guided one-hop propagation to inject graph structural inductive bias. It adopts fixed-budget subgraph sampling and a padding-stacking batching strategy to achieve ID-free and mini-batch efficient training, fully compatible with hardware-optimized attention kernels.

**Compliance With Llm Reviewing Policy:**

Affirmed.

**Key Questions For Authors:**

Though the paper is already well-constructed, several key questions could further deepen the understanding of the proposed work, and the responses may refine our evaluation of the paper:
1.	What are the fundamental reasons for the divergent effects of node features on Planetoid and OGB datasets, and could there be a way to quantitatively characterize their interplay?
2.	Could there be preliminary lightweight strategies to mitigate the linear scaling of compute/memory costs with the number of candidate links?
3.	What targeted optimization methods could be adopted to improve PENCIL’s statistical efficiency on small-scale graph datasets?
4.	The paper has conducted ablation studies on multiplicative residual connections; how could it compare with other residual fusion forms (e.g., additive) in injecting graph inductive bias?

**Limitations:**

yes

**Strengths And Weaknesses:**

Strength
Soundness: The paper demonstrates strong soundness with rigorous theoretical analyses and comprehensive experiments across benchmark datasets, and validates improvements via ablation studies.
Presentation: The paper is well-structured with a clear logical flow from problem formulation to method design and experimental results.
Significance: This paper fills the gap of plain Transformer for large-scale link prediction, proposing an ID-free and hardware-efficient paradigm that addresses the expressivity-scalability trade-off in existing models.
Originality: The work innovates with link-centric subgraph encoding and multiplicative residual connection.

Weakness

1.The model shows low statistical efficiency and inferior performance to some SOTA baselines on small-scale datasets, indicating possible data-hungry issue.
2. The compute and memory costs scale linearly with the number of candidate links; while feasible optimization directions (e.g., subgraph computation caching) are proposed, no in-depth exploration is conducted.

---

> ### Author Rebuttal · Authors · 2026-03-31
>
> We thank you for your constructive feedback. We hope our responses address your concerns and merit a higher score.
>
>
> **[W1, Q3] Low performance on small datasets and solutions**
>
> We have explicitly noted that the absence of hard-coded inductive biases demands a larger volume of data, with supported evidence from (Sanford et al., 2024) and (Dosovitskiy et al., 2021).
>
> A natural solution is pretraining on abundant data, as in text and vision. Applying this to link prediction is non-trivial because graphs differ widely in attributes and structure, and is an active topic in graph foundation models.
>
> **[Q1] On structural-feature-task interplay**
>
> The interplay among node features, structure, and downstream tasks is a longstanding challenge in graph ML. Prior work shows that their usefulness is highly task- and dataset-dependent [1,2]: node features may be redundant with structure, or even harmful when misaligned with the task [2]. Until now, there is no dedicated data-perspective study to quantify the interplay in link prediction, so we opt for feature ablation in our study.
>
> We further demonstrate that the divergent effects of node features are due to data problem not because of the model design. Specifically, we ran a controlled synthetic study on latent-space graphs [3], varying $s$ (structural signal) and $\mu$ (node-feature signal) while keeping graph size and average degree approximately fixed. Detailed distribution of edge types and splits can be found in [the anonymized URL](https://anonymous.4open.science/r/pencil-4239/ls_viz.png).
>
> We study two regimes: varying structural signal $s$ at fixed $\mu = 1$, and varying node-feature signal $\mu$ at fixed $s = 1$, with two edge cases ($s = 2, \mu = 0$) and ($s = 0, \mu = 8$). Larger $s$ and $\mu$ correspond to stronger structural and node-feature signal, respectively. Every model has 2 layers, with the hidden channel fixed to 256. MRR results are displayed under the following tables.
>
> | Model | s0-m1 | s1-m1 | s2-m1 | s2-m0 |
> | --- | --- | --- | --- | --- |
> | CN | 1.58 | 4.78 | 4.48 | 4.48 |
> | AA | 2.85 | 5.43 | 7.37 | 7.37 |
> | GCN | 75.48 | 70.08 | 83.68 | **40.67** |
> | GAT | 71.44 | **81.29** | 82.11 | 6.27 |
> | PENCIL | **77.63** | 74.37 | **93.59** | 36.01 |
>
> | Model | s1-m0 | s1-m2 | s1-m8 | s0-m8 |
> | --- | --- | --- | --- | --- |
> | CN | 4.78 | 4.78 | 4.78 | 1.58 |
> | AA | 5.43 | 5.43 | 5.43 | 2.85 |
> | GCN | 10.09 | 71.04 | 73.85 | 77.05 |
> | GAT | 5.36 | 83.57 | 83.04 | 76.01 |
> | PENCIL | **17.21** | **84.90** | **92.05** | **95.34** |
>
> We can make the following observations. First, pairwise heuristics are weak predictors across scenarios. Second, PENCIL benefits more consistently from stronger node-feature signal than the other baselines. Third, because GAT is strongly feature-dependent, it degrades sharply when features are noisy ($\mu = 0$), whereas PENCIL is much more robust. In general, PENCIL delivers strong performance with help of node features. In other words, we believe this apparent incompatibility is primarily a **data-regime issue rather than a model limitation of PENCIL**.
>
> **[W2, Q2] Scalability**
>
> We explicitly acknowledge this limitation in the main text. The cost of PENCIL arises from edge-centric subgraph sampling used to compute edge representations. In practice, real-world deployment can often be decomposed into offline and online stages. PENCIL is particularly well suited to the offline stage with constrained candidate sets, thereby reducing the burden during online inference. When the remaining online candidate set is small, the cost can be mitigated by distributed inference, although this isn't a sustainable solution. An important direction is therefore to develop a more scalable variant of PENCIL that avoids edge-centric subgraph sampling. More generally, this highlights a no-free-lunch trade-off: PENCIL exchanges computation for bounded memory, whereas embedding-based methods exchange storage for faster inference.
>
> Possible directions include batching multiple link-centered subgraphs jointly or combining retrieval-style scoring with PENCIL to reduce subgraph sampling. We leave these designs for future works, since our work focuses mainly on theoretical properties that allow PENCIL to be comparable with existing link predictors.
>
> **[Q4] Additive Residuals**
>
> Because additive residuals are token-wise, they cannot encode cross-token graph structure the way multiplicative residuals can. Specifically, in Table 9, PENCIL \ MR is the variant where only additive residuals are allowed, with the only input projection layer injecting graph structures. We can observe the degradation in performance when only additive residuals are present.
>
> [1] Oversmoothing, "Oversquashing", Heterophily, Long-Range, and more: Demystifying Common Beliefs in Graph Machine Learning.
>
> [2] No Metric to Rule Them All: Toward Principled Evaluations of Graph-Learning Datasets.
>
> [3] Latent Space Approaches to Social Network Analysis.

---

> > ### Author Rebuttal · Reviewer_yRG3 · 2026-04-04
> >
> > Thank you for addressing my concerns. I will maintain my current score.

---

> > > ### Author Response · Authors · 2026-04-06
> > >
> > > We thank the reviewer for their positive feedback on our paper and rebuttal. We are committed to making the promised revisions in the final version of the manuscript. Please let us know if you have any further questions.

---

### Decision · Program_Chairs · 2026-04-30

**Decision:**

Accept (regular)

**Comment:**

Reviewers found the core idea appealing: a plain Transformer with link-centric subgraph tokenization can be surprisingly competitive for link prediction, and the empirical study and theoretical analysis were viewed as solid. The main concerns were that the paper does not yet extract sufficiently deep takeaways from this result, that some theory-to-experiment links remain incomplete, and that deployment scalability should be discussed more carefully. The rebuttal addressed several questions and improved confidence, but it did not fully resolve the remaining concerns about insight, justification, and practical scope. Overall, I recommend Reject or Weak Accept (if there is room) because the paper is thought-provoking and promising, but the current version does not yet make a strong enough case for a solid acceptance.